# Structural transitions in the stepwise assembly of proteasome core particles

Eric Mark [1,4], Paula C. Ramos[2,4], Maria M. Nunes[2], Ana C. Matias[2,3], R. Jürgen Dohmen [2] ✉ & Petra Wendler [1] ✉

20S catalytic core particles (CP) of eukaryotic 26S proteasomes are composed of two identical halves comprising 14 distinct subunits. 15S precursor complexes (PC) represent detectable half-CPs assembly intermediates lacking the β7-subunit but containing assembly chaperones Ump1 and Pba1-Pba2. Incorporation of β7 drives 15S-PC dimerisation and further CP maturation. Our cryo-EM structures of the yeast 15S-PC and all 13S-PC-derived intermediates suggest that assembly in yeast is not restricted to a single trajectory, but instead involves alternative, and potentially simultaneous pathways. Comparison of the intermediates reveals how Ump1 and β-subunits become structured with each additionally incorporated β-subunit, and how this prepares peptidase sites for auto-activation. We identify two transient interactions of Pba1 with the α-ring, which are important for an ordered progression of maturation. Pba1 loop 81-117 intercalates between subunits α3 and α4 in 13S-15S-PCs and is displaced upon 15S-PC dimerisation. The second interaction involves the α1 N-terminus, deletion of which leads to a defect in Pba1-Pba2 release. These findings indicate how changes in α-ring subunit conformations coordinate CP maturation with Pba1-Pba2 release.

The 26S proteasome is a central and essential component of the ubiquitin/proteasome system in eukaryotic cells[1]. Because selective protein degradation by the proteasome is of critical importance, both in quality control and regulation of proteins, the proteasome is involved in a multitude of cellular processes and has emerged as a drug target in the treatment of certain types of cancer[2]. The eukaryotic proteasome is composed of two 19S regulatory particles (RP) and the 20S catalytic core particle (CP)[3]. CPs are composed of 28 (2×14 distinct) subunits and assemble independently from RPs in a process involving various dedicated chaperones that are unique to eukaryotic proteasome assembly pathways[4]. The central chamber of the CP is formed by two heptameric rings of β-subunits that are sandwiched between the two distal α-rings. The CP harbours the peptide-cleaving catalytic activities, which are mediated by active sites in three of the seven β-subunits (β1, β2, and β5) of each ring[5]. The active sites of each of these subunits are characterized by N-terminal threonine residues, which are exposed after autocatalytic removal of propeptides[6].

In contrast to 13 of the 14 genes encoding CP subunits, the genes encoding assembly chaperones are not essential for viability in yeast[6,7]. Their absence, however, causes distinct defects in proteasome assembly with biochemical and phenotypic consequences. Most dramatic are the defects of yeast cells lacking the assembly chaperone Ump1, which display poor growth and hypersensitivity to proteotoxic stress[8]. Ump1 is found in early assembly intermediates known as 13S precursor complexes (13S-PC) that contain a complete ring of seven α-subunits plus subunits β2, β3, and β4[9–11]. In *ump1Δ* cells, proteasome biogenesis is impaired, leading to CPs with incompletely matured β-subunits and reduced activity[8]. In mammals, UMP1, alias POMP, is essential for cell viability[12]. The 13S-PC, in addition to Ump1 and the above-mentioned α- and β-subunits, contains the heterodimeric

[1]Institute of Biochemistry and Biology, Department of Biochemistry, University of Potsdam, Potsdam-Golm, Germany. [2]Institute for Genetics, Center of Molecular Biosciences, Department of Biology, Faculty of Mathematics and Natural Sciences, University of Cologne, Cologne, Germany. [3]Present address: IPMA—Portuguese Institute for the Sea and Atmosphere, EPPO—Aquaculture Research Station, Olhão, Portugal. [4]These authors contributed equally: Eric Mark, Paula C. Ramos. ✉e-mail: j.dohmen@uni-koeln.de; pewendler@uni-potsdam.de

chaperone Pba1-Pba2, which is bound to the distal α-ring surface[13]. Pba1-Pba2 (or its mammalian ortholog PAC1-PAC2) facilitates precursor assembly and promotes proper proteasome maturation by blocking off-pathway association of α-rings[14–16]. Deletion mutations (*pba1Δ* or *pba2Δ*) cause only mild phenotypes in yeast but display genetic interactions with other mutations affecting proteasome regulation or assembly (*rpn4Δ, ump1Δ, pre9Δ*)[11,15,16]. Lack of Pba1-Pba2 function, however, causes subtle changes in the composition of proteasomal complexes, with an increase in Blm10-associated CPs[17]. In mice, knock-out of the Pba1 ortholog PAC1 is embryonically lethal[18], underlining the importance of this chaperone for proper proteasome biogenesis in higher eukaryotes. During the assembly process, Pba1-Pba2 remains associated with nascent CPs until their final maturation[17,19,20]. The latter process involves removal of β-subunit propeptides and the degradation of Ump1, which becomes encased upon formation of CPs by the β7-driven dimerisation of two so-called 15S-PCs[8,19,21,22]. An additional chaperone heterodimer, Pba3-Pba4, acts in earlier assembly steps preceding formation of 13S-PCs[23,24].

15S-PCs are derived from 13S-PCs by the addition of β1, β5, and β6 in an order that has been still unclear. Previous studies reported 13S-PC structures containing both β5 and β6[13,21], implying that association of β5 and β6 occurs before β1 incorporation (Table S1). The 15S-PC bears all 14 CP subunits but β7[11,25], and its structure has not been solved. Half-proteasomes, consisting of β7-bound 15S-PCs, have been purified from recombinant insect cell systems and human cells (Supplementary Table 1)[21–23,26], but remain undetectable in yeast cells[11,25]. Dimerisation of 15S-PCs by addition of β7 leads to a complete but still immature PC, which we refer to as late-PC[11,19,25]. This complex is characterized by the presence of two Pba1-Pba2 molecules that are attached to α-ring surfaces, two molecules of Ump1 encased within the structure, and unprocessed β-subunits[19,21–23,26]. Conformational changes in the β-rings upon dimerisation trigger autocatalytic processing, probably initiating with β2 activation[19]. Mature β2 promotes further maturation steps by shortening of other propeptides. The active β-subunits then degrade Ump1. These maturation steps not only go along with structural changes in the β-rings but also in the α-rings, which lead to a release of Pba1-Pba2, an indicator of the completion of CP maturation[17,19]. Because CP assembly occurs rather fast, most assembly intermediates, such as the 13S-PC and the late-PC, are very short-lived and therefore present at only very low steady state concentrations in wild-type (WT) cells. Therefore, purification and characterization of such complexes often employ cells with mutations in α- or β-subunits that slow down assembly or maturation steps[13,19,22]. Downsides of these otherwise helpful approaches are that these mutations go along with structural changes. In contrast to the above-mentioned PCs, the 15S-PC is detectable at steady state in WT cells, albeit at relatively low levels[8].

In this work, we have increased the accumulation of PCs by mutating the β7-subunit, which is not a component of 13S- or 15S-PCs, in order to purify such PCs without any mutations in the incorporated α- or β-subunits. This approach allowed us to perform a cryo-EM-based structural characterization of 13S- and 15S-PCs, as well as multiple structurally uncharacterised intermediates between them. This analysis hints for unexpected alternative routes of β-subunit incorporation and reveals a number of previously unrecognised details regarding structural transitions in subunits and chaperones that underlie the stepwise assembly and maturation of CPs.

## Results

### Purification of in vivo-formed authentic CP assembly intermediates from *S. cerevisiae*

Human and yeast PC structures were reported in previous cryo-EM studies, some of which either employed reconstitution of the complexes in insect cells or utilized yeast strains with mutated β-subunits, respectively[13,21,23,26]. These approaches revealed early assembly intermediates such as the 13S-PC and a 13S-PC associated with β5 and β6

(Supplementary Table 1), but they could not resolve the structure of a fully assembled 15S-PC. In addition, due to the usage of β1 mutant strains, the timing of β1 incorporation during assembly of yeast CPs remained unclear. We set out to accumulate native early-PCs in yeast by using a strain lacking a C-terminal extension of the β7 subunit (β7-ΔCTE)[27]. Absence of the β7-CTE significantly impairs dimerisation of 15S-PCs, resulting in their accumulation. Because β7 is not a part of 15S-PCs or earlier assembly intermediates, C-terminal truncation of β7 has no impact on the structures of these complexes. One consequence of the accumulation of 15S-PCs, however, is a shortage of the Pba1-Pba2 chaperone, because it is only recycled when the assembly pathway reaches the final steps of CP maturation[20]. Consequently, Pba1-Pba2-deficient α-ring surfaces associate with Blm10. To compensate, we concomitantly overexpressed Pba1 and Pba2 in a *blm10Δ* background[20] thereby enriching authentic Pba1-Pba2-capped 15S-PCs and earlier intermediates, which facilitated their affinity purification via N-terminally FLAG-6xHis-tagged Ump1 (Supplementary Fig. 1).

This strategy yielded sufficient material to determine high-resolution cryo-EM structures of six early PCs ranging from 13S-PC to 15S-PC, differing primarily in the β-ring composition (Fig. 1, Supplementary Fig. 2, Table 1). The 13S-PC structure, resolved at 3.3 Å resolution (Supplementary Fig. 3A), comprises a fully assembled α-ring, β2-4, and the assembly chaperones Pba1-Pba2 and Ump1. It overlays well with the structure of a yeast 13S-PC isolated from a *pre3-1* strain (β1-G34D) (PDB 7LSX, $C_\alpha$ rmsd 1.03 Å, over the entire structure)[13]. As the 13S-PC represents the earliest of the structurally characterised intermediates, all later intermediates contain all of its subunits. Unlike prior studies, we captured a 13S-PC with an additional β1-subunit, termed 13S + β1-PC, at 3.2 Å (Supplementary Fig. 3B), and identified 13S-PCs containing the subunits β1-5 (13S + β1 + β5-PC) at 4.2 Å (Supplementary Fig. 3C), modelled as a $C_\alpha$ structure due to low resolution of the 3D reconstruction. A previously uncharacterized 13S + β5-PC, containing β2-5, was found in a 13S-13S + β5 dimer, in which a 13S-PC and a 13S + β5-PC are wedged together (Supplementary Fig. 4A). As for a complex that we termed the 13S-15S dimer (Supplementary Fig. 4B), the 13S-13S + β5 dimer was exclusively detected in FLAG-elutions after centrifugal filtration. Both complexes dissociate upon dilution before grid preparation, indicating that they are non-physiological artefacts arising from the high complex concentration needed for cryo-EM sample preparation. The 13S-13S + β5 dimer was resolved to 3.1 Å. In the 13S-13S + β5 dimer, both halves associate to form an artificial antiparallel β2-β2′ interface, positioning the β4 and β5′ subunits close but too far apart for interaction (Supplementary Fig. 5). The β2-subunits are hydrogen-bonded by a parallel β-sheet (encompassing β2 and β2′ residues 118-121), while β4 and β5′ lack contact. Intriguingly, these findings explain previously detected long-distance cross-links in our study involving early PCs from the same genetic background (Supplementary Table 2)[20]. Like earlier studies involving yeast and human PCs[13,20,21,26], we identified a PC species containing β2-6 (13S + β5 + β6-PC), resolved at 3.3 Å (Supplementary Fig. 3D), which closely matches a recently published structure from *pre3-1/β1* background (PDB 7LS6, rmsd 1.00 Å over the entire structure)[13]. Finally, we obtained a 3.1 Å reconstruction of the native 15S-PC, comprising β1-6 (Supplementary Fig. 3E). Among all intermediates in our dataset, the 15S-PC accounts for 13.7% of particles, substantially less than the 13S-PC (37%) and 13S + β5-PC (25.6%), but almost twice that of all other complexes (Fig. 1).

### β-subunit incorporation during assembly of 15S-PCs does not follow one fixed order

Neither recent cryo-EM studies on human complexes[21,23,26], nor a study on PCs isolated from yeast cells bearing the *pre3-1* mutation[13] detected any 13S + β1-PC species (Supplementary Table 1). These observations suggested that incorporation of β5 and β6 takes place before that of β1. Our structures of the 13S-, 13S + β5 + β6- and 15S-PCs as well as the

α1 α2 α3 α4 α5 α6 α7 β1 β2 β3 β4 β5 β6 Pba1 Pba2 β2pro β5pro FLAG-6xHis-Ump1

| complex | 13S-PC | 13S+β1-PC | 13S+β5-PC | 13S+β1+β5-PC | 13S+β5+β6-PC | 15S-PC |
|---|---|---|---|---|---|---|
| rel. amount [%] | 37.0 | 8.7 | 25.6 | 8.2 | 6.8 | 13.7 |
| global resolution [Å] | 3.3 | 3.2 | 3.1 | 4.2 | 3.3 | 3.1 |
| resolved residues | | | | | | |
| Ump1 | 32-42 48-127 131-148 | 31-42 48-126 133-148 | 21-126 131-148 | 32-124 131-148 | 21-125 132-148 | 21-125 132-148 |
| Pba1 | ✓ | ✓ | ✓ | ✓ | ✓ | ✓ |
| Pba2 | ✓ | ✓ | ✓ | ✓ | ✓ | ✓ |
| β1pro | / | / | / | / | / | / |
| β1 | / | 1-20 32-165 173-190 | / | 6-17 32-166 173-190 | / | 1-17 32-166 173-190 |
| β2pro | ✓ | ✓ | ✓ | ✓ | ✓ | ✓ |
| β2 | 1-20 32-165 170-192 216-231 | 1-20 32-192 216-231 | 1-19 32-192 216-231 | 1-20 34-192 216-231 | 1-19 32-165 169-192 216-231 | 1-20 32-192 216-231 |
| β3 | 4-29 39-203 | 4-29 39-203 | 4-29 39-175 180-202 | 7-29 42-203 | 5-29 39-203 | 4-29 39-203 |
| β4 | 1-20 31-196 | 1-21 31-196 | ✓ | 1-17 32-196 | ✓ | ✓ |
| β5pro | / | / | (-75) - (-67) | (-75) - (-66) (-4) - (-1) | (-75) - (-66) (-4) - (-1) | (-75) - (-66) (-4) - (-1) |
| β5 | / | / | 1-19 35-139 143-165 174-189 | 1-19 36-43 51-165 174-190 | 1-165 174-190 | 1-165 174-190 |
| β6pro | / | / | / | / | / | / |
| β6 | / | / | / | / | 10-25 41-149 175-193 202-217 | 8-26 41-147 173-193 202-217 |
| β7 | / | / | / | / | / | / |

**Fig. 1 | Cryo-EM maps of early proteasomal PCs.** Structures of six distinct early PC complexes isolated from a β7-ΔCTE strain. Subunits, chaperones, and resolved regions present in the maps are indicated by check marks or numbers of the resolved residues. Slashes indicate absent subunits. The 13S + β5-PC was only found in artificial 13S-13S + β5 dimers. The colour code for all subunits present is given above.

**Table 1 | Data collection and model-building statistics**

| | 13S-PC | 13S + β1-PC | 13S + β1 + β5-PC | 13S + β5 + β6-PC | 15S-PC | 13S-13S + β5 dimer | 13S-15S dimer |
|---|---|---|---|---|---|---|---|
| EMDB | 54029 | 54032 | 54048 | 54047 | 54046 | 54045 | / |
| PDB | 9RL3 | 9RLA | 9RM1 | 9RMO | 9RLZ | 9RLT | / |
| **Data collection** | | | | | | | |
| Magnification | | | | 105,000 | | | |
| Voltage [kV] | | | | 300 | | | |
| Electron exposure [e⁻/Å²] | | | | 64.64 | | | |
| Defocus range [μm] | | | | 0.8–2.8 | | | |
| Pixel size [Å] | | | | 0.834 | | | |
| **Processing** | | | | | | | |
| Initial particle images [no.] | | | | 5,283,715 | | | |
| Final particle images [no.] | 40,516 | 53,118 | 50,233 | 41,532 | 54,528 | 156,855 | 29,169 |
| Symmetry imposed | C1 | C1 | C1 | C1 | C1 | C1 | C1 |
| Map resolution [Å] | 3.31 | 3.16 | 4.11 | 3.29 | 3.12 | 3.09 | 3.65 |
| FSC threshold | 0.143 | 0.143 | 0.143 | 0.143 | 0.143 | 0.143 | 0.143 |
| **Refinement** | | | | | | | |
| Initial model used [PDB code] | 8RVL | 8RVL | 8RVL | 8RVL | 8RVL | 8RVL | / |
| Map sharpening B-factor [Å²] | 39.3 | 42.4 | 54.7 | 40.9 | 41.4 | 61.6 | / |
| Model composition: | | | | | | | |
| Non-hydrogen atoms | 23,220 | 24,539 | 13,220 | 26,081 | 27,516 | 47,532 | / |
| Protein residues | 2980 | 3151 | 3305 | 3356 | 3536 | 6098 | / |
| Ligands | 0 | 0 | 0 | 0 | 0 | 0 | / |
| R. m. s. deviations: | | | | | | | |
| Bond lengths [Å] | 0.002 | 0.002 | 0.001 | 0.002 | 0.003 | 0.003 | / |
| Bond angles [°] | 0.415 | 0.428 | 0.353 | 0.402 | 0.421 | 0.440 | / |
| Validation: | | | | | | | |
| MolProbity score | 1.97 | 1.56 | 0.78 | 1.30 | 1.57 | 1.64 | / |
| Clashscore | 5.43 | 3.67 | 0.36 | 4.38 | 4.86 | 3.47 | / |
| Poor rotamers [%] | 3.17 | 1.54 | 0.00 | 1.21 | 1.78 | 2.38 | / |
| Ramachandran plot: | | | | | | | |
| Disallowed [%] | 0.10 | 0.06 | 0.03 | 0.00 | 0.00 | 0.03 | / |
| Allowed [%] | 4.16 | 3.77 | 2.83 | 2.09 | 2.67 | 3.30 | / |
| Favoured [%] | 95.74 | 96.17 | 97.14 | 97.91 | 97.33 | 96.67 | / |

13S + β5-PC in the 13S-13S + β5 dimer support this order of events. However, we also obtained structures of a 13S + β1-PC and a 13S + β1 + β5-PC (Fig. 1) suggesting an additional, alternative assembly pathway, in which β1 is incorporated before β5 and β6 (Fig. 2A).

An alternative possibility is that the observed complexes might have arisen not as genuine assembly intermediates but as dissociation products formed during purification. To test whether β1 can assemble with 13S-PCs before β5 and β6 in vivo, we generated strains that allowed to selectively shut down expression of β5 or β6 by placing the galactose-inducible and glucose-repressible $P_{GAL1}$ promoter in front of the endogenous gene copies of the respective genes (*PRE2* and *PRE7*)[28]. To verify efficient shutdown of β5 and to follow incorporation of this subunit into PCs, the 3' end of the *PRE2* gene was, in addition, fused to a sequence encoding a C-terminal 2xHA-tag to enable detection of the β5 subunit with high sensitivity. Upon shift to glucose media, β5-2xHA was undetectable in a native-PAGE/Western blot analysis of proteasomal complexes in extracts of the β5-shutdown (SD) strain (Supplementary Fig. 6). Simultaneous probing of the same western blot membrane for the presence of α7 showed that β5-SD caused a drastic reduction in 20S CP formation with a simultaneous accumulation of early PCs. The latter effect was similarly observed for the β6-SD strains, which is consistent with the essential role of both subunits for the completion of CP assembly. To assess β1 incorporation, we introduced an N-terminal FLAG-tag upstream of the β1 propeptide (β1pro),

enabling selective pull-down of β1-containing PCs. In contrast to an untagged control, FLAG-tagged β1 pulled down 13S + β1-PCs without β5 (upon β5-SD), and 13S + β1 + β5-PCs (upon β6-SD), thereby proving that neither incorporation of β5 nor of β6 is a requirement for β1 assembly with 13S-PCs (Fig. 2B). These in vivo data support our structural observations and indicate an alternative assembly pathway in which β1 assembles with 13S-PCs before β5 and β6 (Fig. 2A). However, the higher abundance of β5-containing versus β1-containing complexes in our EM dataset (Fig. 1), suggests that β5 incorporation before β1 is the predominant assembly route in yeast cells (bottom pathway in Fig. 2A).

**Stepwise restructuring of S3/S4 β-hairpin loops prepares nascent CPs for activation**

In the mature CP structure, S3/S4 β-hairpin loops of all β-subunits are located at the contact surfaces with their right-hand neighbours within the same β-ring and are juxtaposed to subunits in the opposing β-ring[5,19]. These structural features may help to stabilise nascent assembly intermediates by promoting the binding of incoming subunits and fixing them in the correct position by a restructuring event[21]. Asp[17] and Lys[29] of the catalytic subunits β1, β2, and β5, which are part of the catalytic triad, are located in the S3/S4 loops, positioning these residues in close proximity to the third residue of the triad, the catalytic Thr[1]. This structural arrangement indicates that the S3/S4 loops

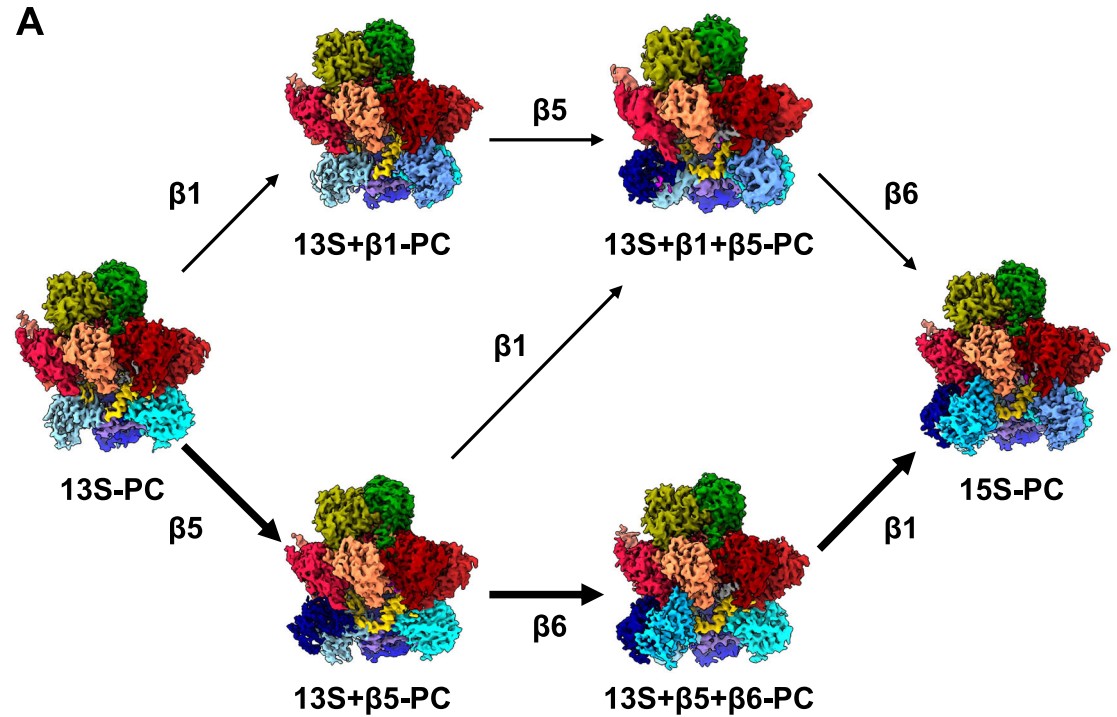

**A**

α1 α2 α3 α4 α5 α6 α7 β1 β2 β3 β4 β5 β6 Pba1 Pba2 β2pro β5pro FLAG-6xHis-Ump1

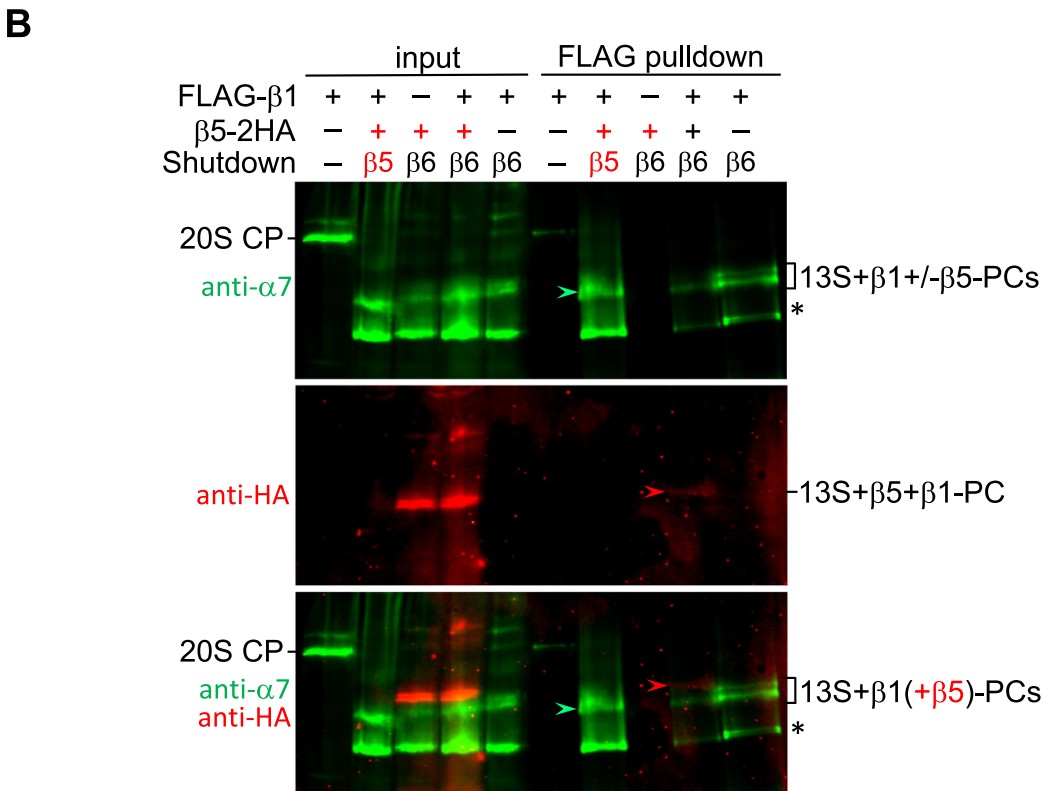

**B**

have an important function in properly positioning the catalytic residues for autocatalytic activation.

The early PC structures presented here allowed us to analyse the roles of all S3/S4 loops without interfering mutations. In the 13S + β5 + β6- and 15S-PCs, the S3/S4 β-hairpin loops of most β subunits appear to be flexible as they are not resolved (Supplementary Table 3). Only the S3/S4 loops of β4 and β5 are resolved in these PCs

(Supplementary Figs. 7 and 8). Residues 25–28 of β4 and β5, which later form sheet S4 in the mature CP, align with β-sheets of their neighbouring subunits in an antiparallel orientation. Specifically, residues 25-28 in β4 align with sheet S10 in β5 (residues 125–129), and residues 25-28 in β5 align with sheet S9 in β6 (residues 135-138) (Supplementary Figs. 7 and 8). In contrast, this alignment is not detectable in the 13S + β1 + β5-PC, indicating that β6 incorporation promotes this

**Fig. 2 | 15S-PC assembly pathways in *S. cerevisiae*. A** Cryo-EM structures solved in this study arranged in possible alternative 15S-PC assembly pathways. Structures are shown as side views. The colour code for all subunits present is given below. The 13S + β5-PC was only found in artificial 13S-13S + β5 dimers. **B** Native PAGE analysis of proteasomal complexes containing FLAG-β1 after down-regulation of β5 or β6 expression and FLAG pulldown. PCs present in extracts from yeast strains expressing the indicated tagged β-subunits either from their endogenous native promoters or, where indicated, from the galactose-induced or glucose-repressed P$_{GAL1}$ promoter, were analysed by native PAGE and western blotting. Cells were grown in selective galactose media or shifted to glucose media for 15 hours. Left part, crude

extract (5% input relative to FLAG pulldown). Right part, FLAG-pulldown (95% of starting material). Electrophoresis was performed in 1 mm 3-12% native PAA gels. Blots were simultaneously probed with polyclonal rabbit anti-α7 and rat anti-HA primary antibodies, and fluorophore-coupled anti-rabbit (green) and anti-rat (red) secondary antibodies. The upper panel shows the green fluorescence signals of the scanned blot, the middle panel shows the red fluorescence signals, and the bottom panel shows the overlay of signals obtained in both channels of the scan. Positions of the 20S CP and distinct 13S+ PCs are indicated. An asterisk marks the band of an earlier PC. Source data are provided as a Source Data file.

structural transition. The S4 strands of β4 and β5 furthermore interact with helices H3 of β5 and β6, respectively, pulling them closer to the core of these subunits when the S3/S4 loops restructure during late assembly stages. For β4, the position of the S3/S4 loop overlays well with that in the *pre1-1*late-PC[19], while the loop in β5 is up to 4.7 Å offset towards the β5 core in the 15S-PC to accommodate the different orientation of β6.

The S3/S4 loops of β1, β2 and β3 remain unstructured in the 15S-PC, most likely due to insufficient conformational constraint. While β1pro and the β1 S3/S4 loop are flexible in all early β1-containing PCs (Figs. 1, 3, and Supplementary Fig. 9), their folding is only observed after β7 incorporation and subsequent dimerisation. Similarly, β2 S3/S4 loop folding is primarily driven by dimerisation. However, the β2 propeptide (β2pro) remains attached to Thr[1] in early precursors (Fig. 1), its cleavage in the *pre1-1*late-PC coincides with S3/S4 loop formation (Fig. 3 and Supplementary Fig. 9). This indicates that dimerisation is the key trigger for autoactivation of β2. In the absence of its direct neighbour (β7), the S3/S4 loop in β6 remains flexible across all structures analysed here. These observations reveal that β-hairpin loop restructuring, active site residue positioning, propeptide cleavage, and CP maturation are tightly coupled. This is expected, as the formation of hydrogen-bond networks - specifically between Lys[29], Thr[1], Ser[129] and Asp[166] in β1, and Lys[29], Thr[1], Ser[131] and Asp[168] in β5 - is essential for the autocatalytic activation of these proteolytic subunits[29]. For autoactivation, these residues must be in close proximity and correctly oriented. In the 15S-PC, only the β2 catalytic site is sufficiently formed for all involved residues to show C$_\alpha$-backbone density (Fig. 3). In contrast, β1 and β5 have flexible loops between residues 165-173 (Supplementary Table 4), leaving Asp[166]/ Asp[168] mispositioned to protonate Thr[1] via Ser[129]/Ser[131] for autoactivation[29]. The β2 loop is stabilised upon β1-association, as it is resolved in all β1-containing early PCs. Intriguingly, it is also folded in both halves of the 13S-13S + β5 dimer, suggesting that interactions with a binding partner in the position of β1 are crucial for its folding.

Together, our findings suggest that the S3/S4 β-hairpin loops and the loops spanning residues 166-173 in the catalytic β-subunits progressively fold during 15S-PC assembly, as conformational freedom is restricted by neighbouring subunits in the same or opposite β-ring following half-CP dimerisation.

**Progressive Ump1 (re)structuring during PC assembly coincides with successive β-subunit addition and dimerisation**
Isolated Ump1 is intrinsically unstructured[30,31]. Its complete structure was only resolved in late-PCs containing all proteasomal subunits and associated chaperones[19,21,32]. The structures presented here emphasize that Ump1 progressively folds upon interaction with proteasomal subunits. In the 13S-PC, residues 1-31, 43-47, and 128-130 of the 148 amino acid long protein are unfolded (Fig. 1 and Supplementary Table 5). Addition of β1 to the PC does not improve folding of Ump1, since helices H5 and H6 of Ump1 are already in close contact with α7 and α1 in the area where β1 joins the complex (Fig. 4A). Binding of β5, on the other side, promotes folding of residues 43-47 into helix H3 of Ump1. This interaction is stabilised by contacts between Ump1-Leu[34]

and the backbone of β5-Gln[89], as well as Ump1-Ser[35] and the backbone of β5-Lys[91] in the 15S-PC (Fig. 4B). When comparing 13S + β1 + β5-PC with 15S-PC or 13S + β5 + β6-PC, it seems that residues 21-31 fold upon addition of β6 to the complex, although β6 does not directly interact with Ump1. We and others have previously shown that the β5 propeptide (β5pro) interacts with the N-terminus of Ump1 and with β6 in the late-PC[19,32]. Hence, folding of residues 21–31 of Ump1 in the PCs might depend on interactions between β6 and β5pro, which should already be in close proximity in these complexes (Fig. 4C). However, residues 21-31 of Ump1 are already folded in the 13S-13S + β5-PC dimer, despite the absence of β6, indicating that folding of this region is caused by restriction of available space rather than an interaction between specific subunits. Structural overlay of the 15S-PC with the *pre1-1*late-PC reveals several interesting details. Firstly, incorporation of β7 into the nascent proteasome pushes subunits β1 and β6 up to 10 Å out of their position in the 15S-PC towards β2 and β5, respectively (Fig. 4C). At the same time, the N-terminus of Ump1 folds due to spatial constraints imposed by β7, β1, and β5pro, as seen in our study of the *pre1-1*late-PC[19]. Secondly, we observe a small (~2 Å) compression of Ump1 along the central axis of the complex (Fig. 4C), possibly induced by the dimerisation and spatial restraints in the newly formed antechamber. This compression goes along with a shift of β2pro towards Ump1 and an upward movement of α1. Thirdly, this restructuring of Ump1 leads to a loss of contact between Ump1-Asp[136] and α3-Arg[114] (Fig. 4B).

We conclude that β7-driven 15S-PC dimerisation enables folding of the Ump1 N-terminus, restricting the available space in the antechamber in such a way that initial Ump1 interactions with the 20S CP subunits are abolished. These changes also induce autoactivation of β2 and β5 as highlighted in previous studies[19,21].

**Pba1 loop inserts between α3 and α4 in early PCs in a similar way as Rpt2 in active 26S proteasomes**
So far, three important interactions between Pba1-Pba2 and α-subunits have been described. First, the C-terminal HbYX motif of Pba1 inserts into the α5/α6 pocket. Second, the HbYX motif at the C-terminus of Pba2 inserts into the α6/α7 pocket[16,33]. Third, the N-terminus of Pba1 extends through the α-ring pore, interacting with all α-subunits as well as Ump1[13].

The early PC structures presented here reveal an additional interaction between Pba1 residues 93-109 and the α3/α4 pocket (Fig. 5A). Human PAC1 lacks this loop; instead, its N-terminus extends into a U-turn directed towards the α3/α4 pocket[21]. There are several hydrogen bonds between this Pba1 loop and the α3/α4 pocket. The Pba1 residues Ser[93] and Asn[99] form hydrogen bonds with α3-Ser[31]/ Glu[27] and α3-Ser[154], respectively. In addition, a network of hydrogen bonds is established between Arg[30] and Asn[79] of α4 and Pba1-Glu[103] (Fig. 5A). The specificity and importance of these interactions is highlighted by the fact that Pba1-Pba2 is unable to associate with PCs in a strain lacking α3 (*pre9Δ*), where a second α4 subunit is incorporated into the CP in place of the missing α3 (Supplementary Fig. 10). Modelling of this Pba1 loop into the α4/α4 pocket of the *pre9Δ*CP[34] indicates spatial and electrostatic hindrance of binding. Intriguingly, the side chain of Pba1-Glu[103] roughly occupies the space

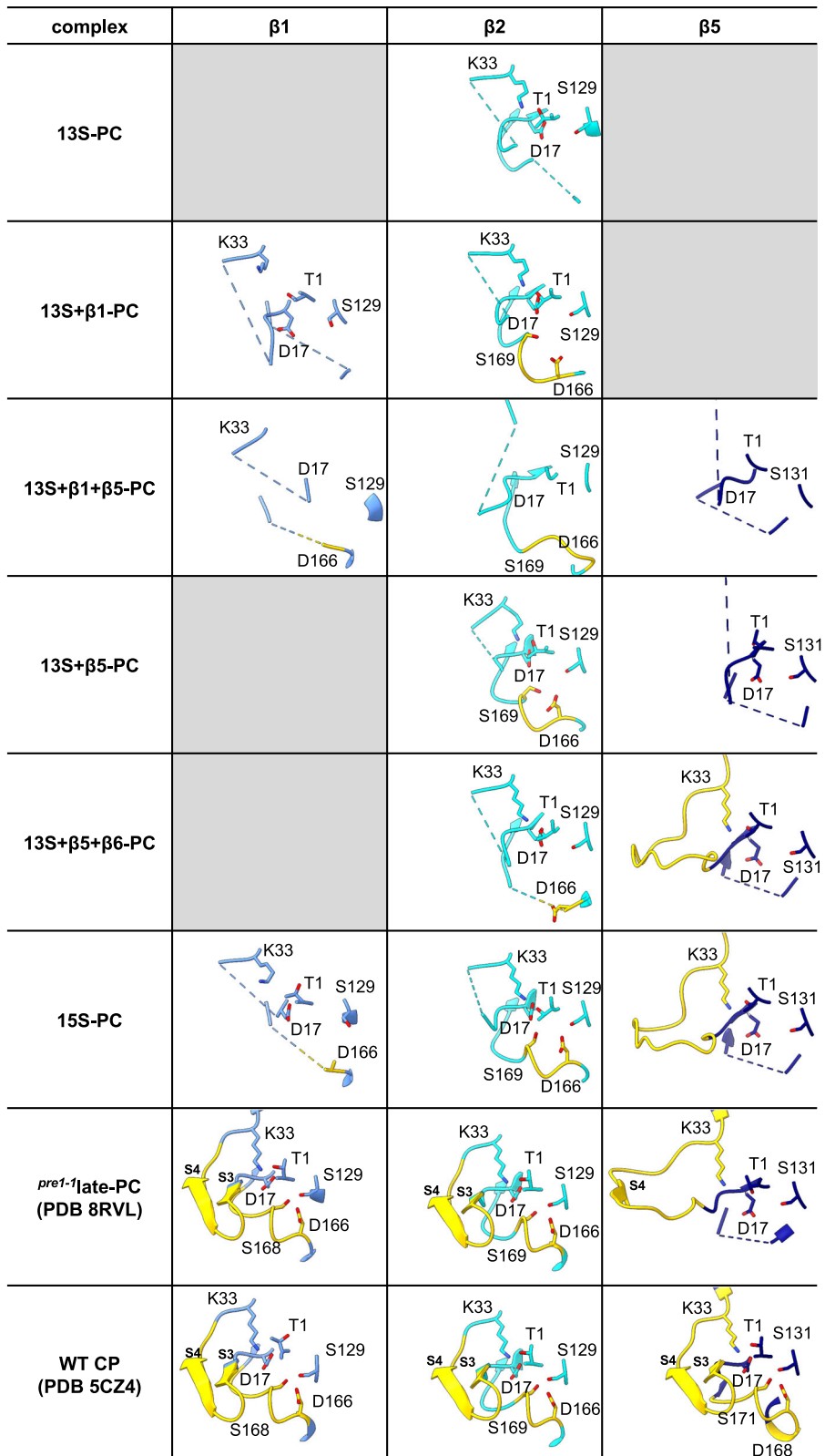

**Fig. 3 | Stepwise structuring of the β1, β2, and β5 catalytic sites during 20S CP assembly.** Structural representation of the catalytic site residues and associated loops in all PCs presented in this work. The catalytic sites of the *pre1-1*late-PC (PDB 8RVL) and WT CP (PDB 5CZ4) are given as references. If resolved, important catalytic site residues are labelled and shown in stick representation. The S3/S4 hairpin loop and the loop region between residues 165 and 173 are shown in yellow. Unresolved loops are shown as dotted lines. The respective subunits of all complexes are superimposed on the left subunit of each panel. The 13S + β5-PC was only found in artificial 13S-13S + β5 dimers.

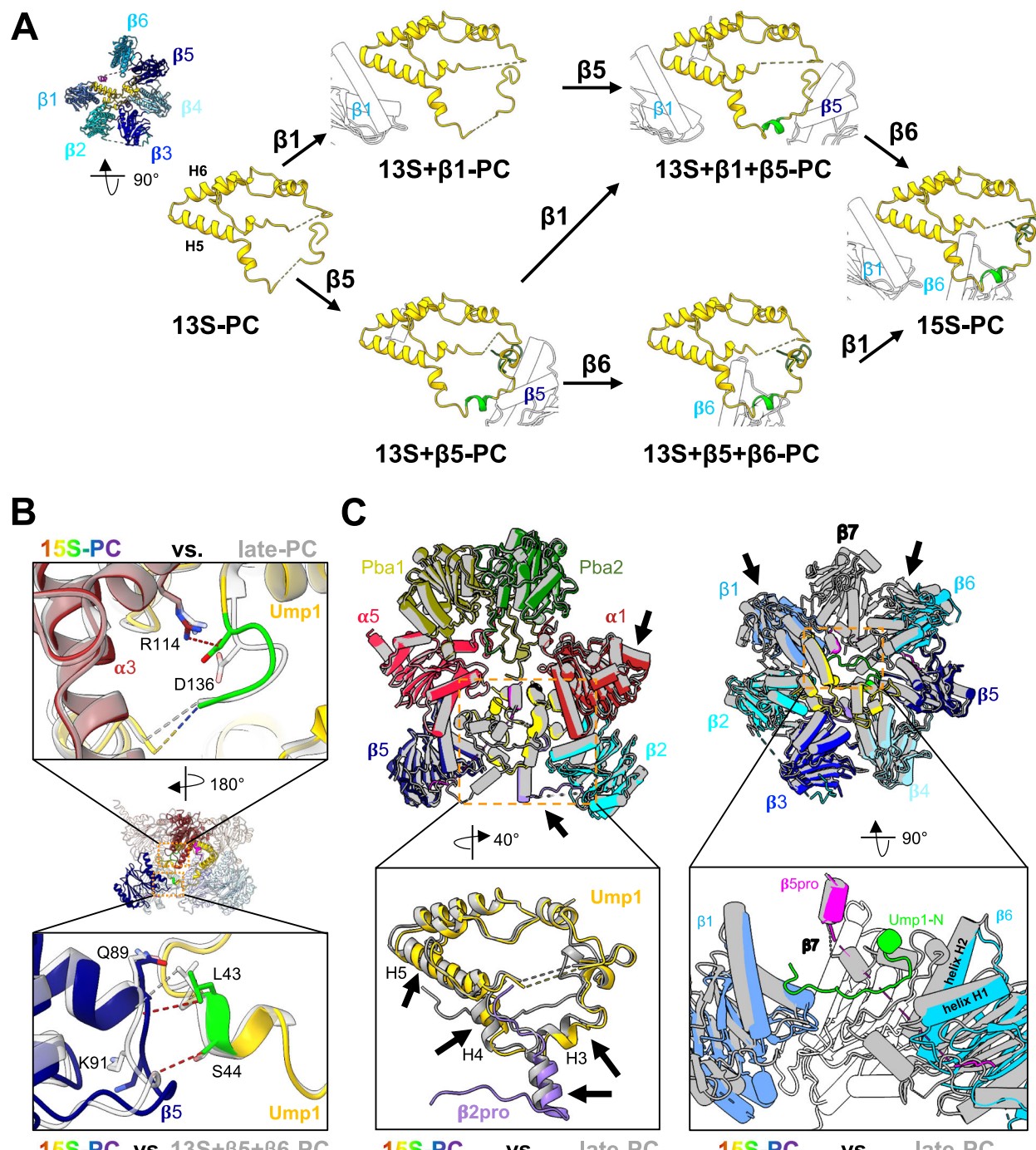

**Fig. 4 | Stepwise (re-)structuring of Ump1 during precursor assembly.**
**A** Ump1 structures as resolved in the early PCs are shown in side view according to the overview of the β-ring in the 15S-PC provided in the upper left after a 90° rotation. Neighbouring β-subunits are indicated as silhouettes, unfolded loops are shown as dotted lines. Helix H3 of Ump1 is highlighted in lime. Residues 21 to 31 of Ump1 are highlighted in dark green. The Ump1 helices H5 and H6 are labelled on the 13S-PC structure. Note that the 13S + β5-PC was only found in artificial 13S-13S + β5 dimers, which appears to be the reason for Ump1 residues 21-31 to be structured even though β6 is not yet incorporated. **B** Shown are details of the interactions

between Ump1 and α3 (top) in the 15S-PC (coloured) compared to the *preI-1*late-PC (grey; PDB 8RVL), and between Ump1 and β5 (bottom) in the 15S-PC (coloured) compared to the 13S + β5 + β6-PC (grey). **C** Superposition of 15S-PC (coloured) onto the α-ring of the *preI-1*late-PC (grey) structure shown from the side displaying only Pba1-Pba2, Ump1, α1, α5, and β2, β5 for clarity (left), or showing only the β-ring and Ump1 (right) from the top. Insets show the isolated view of Ump1 and β2pro (left), and the shifts in β1 and β6 induced by β7 incorporation and dimerisation (right). The location of the Ump1 N-terminus in the *preI-1*late-PC is highlighted in lime. Arrows indicate areas of significant deviation between structures.

of Tyr[436] in the HbYX motif of Rpt2 in the substrate-engaged yeast 26S proteasome (Fig. 5B)[35]. It also interacts with α4-Arg[30]. Thereby, the Pba1 loop intercalates between α3 and α4 in a similar way as the Rpt2 HbYX motif in the 26S proteasome during substrate processing, creating a gap between α3 and α4 (Fig. 5C, D). In comparison with the

position in the mature CP, the Pba1 loop in the 15S-PC keeps α4 from reaching its final position in the α-ring. Notably, the C-terminal regions of Pba1 and Pba2 in the 15S-PC align well with the *preI-1*late-PC structure, but differ from the conformation observed in a reconstituted Pba1-Pba2-20S structure[33] (Supplementary Fig. 11),

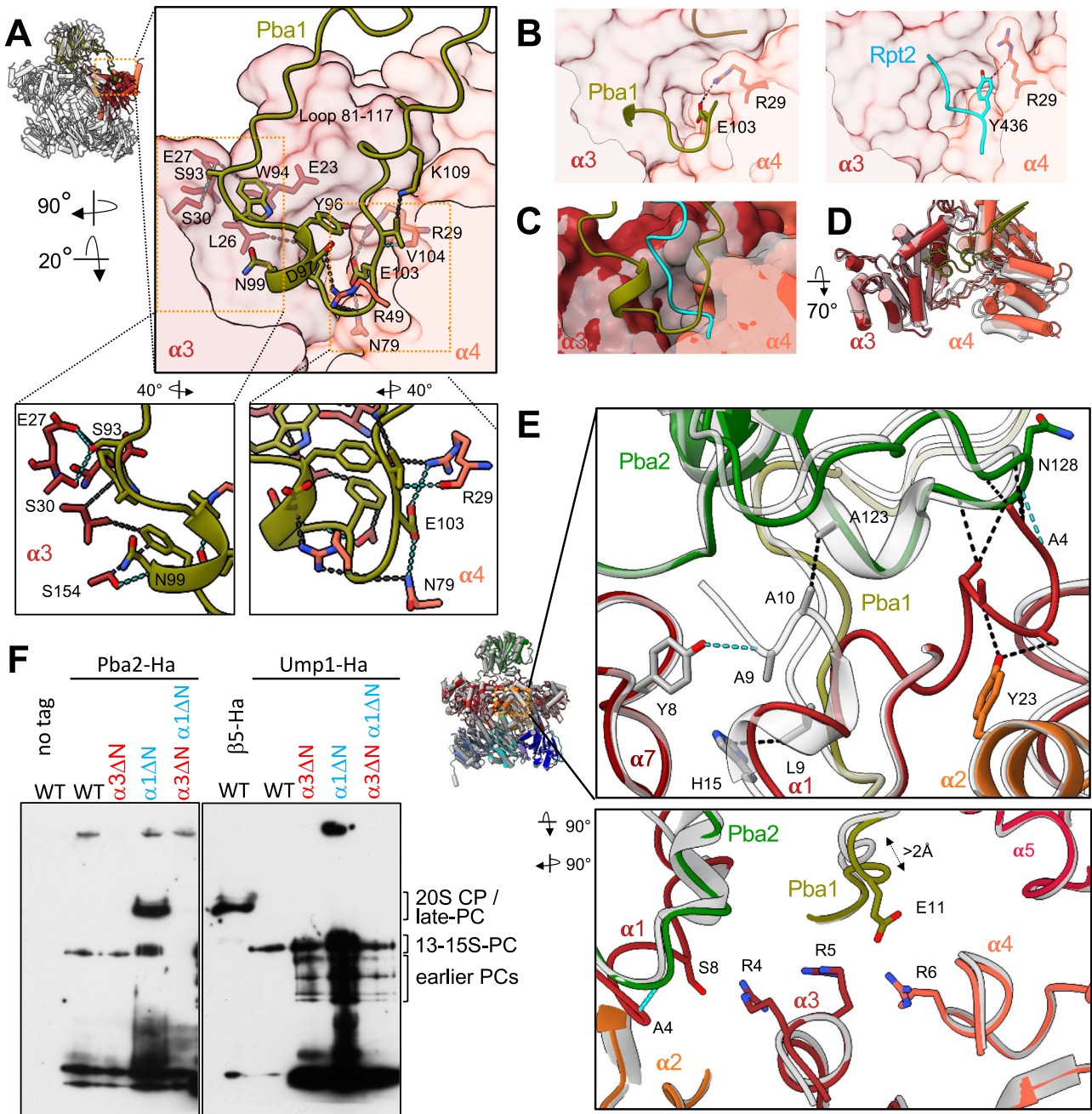

**Fig. 5 | Pba1-Pba2 interactions with 13S to 15S-PCs. A** Interactions of Pba1 residues 93-109 with the α3/α4 pocket of all early PCs, exemplarily shown for the 15S-PC. Only Pba1, α3 and α4 are coloured as indicated in Fig. 2A. Interactions are shown as dashed lines, with hydrogen bonds coloured in cyan. **B** Comparison of Pba1$^{93-109}$ and Rpt2 binding to the α3/α4 pocket in the 15S-PC (left panel) and substrate-engaged 26S proteasome (PDB 6EF3) (right panel). The interaction between α4-Arg$^{29}$ and Pba1-Glu$^{103}$ and Rpt2-Tyr$^{436}$ of its HbYX motif are shown in the same viewing direction as in (**A**). **C** Superposition of the α3/α4 pocket in the 15S-PC (coloured) and 20S CP (grey; PDB 5CZ4) on α3 (left). The α3/α4 pocket is shown in surface depiction. The Pba1 loop in the 15S-PC is shown in cartoon representation, coloured and aligned as in (**B**). Rpt2 HbYX can penetrate the α3/α4 pocket of the closed 20S CP, while the Pba1 loop is sterically hindered. **D** Top view onto superimposed α3 and α4 of the 15S-PC with intercalated Pba1 loop as depicted in (**C**) and 20S CP. Subunits are shown as a tubed cartoon

representation. **E** Comparison of interactions of the α1-N-terminus with surrounding subunits in the 15S-PC (coloured) and $^{pre1-1}$late-PC (grey; PDB 8RVL) shown from the side (top) and as a top view (bottom). Interactions are shown as dashed lines, and hydrogen bonds are coloured in cyan. **F** Analysis of proteasomal complexes in strains with N-terminal truncations (ΔN) of α1 (2-16) and α3 (2-10). Complexes in extracts from WT and strains with the indicated truncations carrying C-terminal 2xHA-tags on either Pba2 or Ump1 were analysed by native PAGE and anti-HA western blotting with ECL detection. A WT strain without an HA-tag served as a control for the specificity of antibody detection. A WT strain with an HA-tag on β5 served as a size marker for the 20S CP. Positions of 20S CP in the WT, late-PC with Pba2-HA in α1-ΔN, as well as of 13-15S-PCs or earlier PCs, are indicated. Source data are provided as a Source Data file. The experiments were repeated once with similar results.

supporting our recent hypothesis that Pba1-Pba2 adopt distinct binding modes in PCs compared to mature CPs[19].

### The N-terminus of α1 is involved in a signaling cascade to dissociate Pba1-Pba2 from proteasomal PCs

Within all the early PCs in this study, the α1 N-terminal residues 3-16 interact with Pba1, Pba2, α2, and α3 (Fig. 5E and Supplementary Fig. 12A). In the [pre1-1]late-PC[19], however, the N-terminal residues of α1 appear rearranged after 15S-PC dimerisation, altering several interactions between α1 and its contacting subunits. While this relocation disrupts interactions between α1 and α2 as well as Pba2, it creates new contacts between α1 and the N-terminus of α7. Intriguingly, this movement is accompanied by a more than 2 Å shift of the Pba1 N-terminus in the pore towards α7 and a small shift of the α3 N-terminus towards α2. Although the resolution of our maps does not allow for exact interpretation of side chain locations in this area, we hypothesize that relocation of the α1 N-terminus might destabilise the α3 N-terminus. The α3 N-terminus, in turn, needs to occupy a space that is taken by the Pba1 N-terminus in the PCs to close the α-ring pore in the mature CP. We speculate that the signal of relocation could be transmitted via α1-Ser[8], α3-Arg[4] and -Arg[5], α4-Arg[6] to Pba1-Glu[11] (Fig. 5E), which is the residue that appears to cause the shift of Pba1 through interactions with α3 and α4. This structural transition might prepare for the pulling back of the Pba1 N-terminus and closure of the α-ring gate by the α-subunits' N-termini.

Because of this striking relocation of the α1 N-terminus, we asked what the effects of a deletion of this peptide would be on CP assembly and maturation. To this end, we generated yeast mutants expressing α1/Scl1 lacking residues 2-16 (Scl1-Δ2-16). Two such mutant alleles were stably introduced into the genome of otherwise WT strains expressing epitope-tagged versions of either Pba2 or Ump1. Native gel analyses of the proteasomal complexes present in these mutants showed that the deletion of the α1 N-terminus causes a drastic accumulation of larger assembly products, which appeared to be late-PCs still associated with the Pba1-Pba2 chaperone, as indicated by the presence of the Pba2-HA-tag (Supplementary Fig. 12B). In contrast to the [pre1-1]late-PC, this [scl1-Δ2-16]late-PC does not contain detectable Ump1, indicating that the α1/scl1-Δ2-16 mutation does not interfere with degradation of this chaperone. We conclude that deletion of the α1 N-terminus causes a CP maturation defect hindering the release of Pba2-HA. Our structural analysis suggests that the N-terminus of α1, which is in bonding distance to the loop segment encompassing residues 121–138 of Pba2 and the N-termini of α2 and α3, prevents the latter from prematurely closing the central pore in the α-ring, which apparently would result in trapping of the Pba1-N-terminus. To test this interpretation, we have compared a set of strains, in which the N-termini of α1 and α3 were truncated either alone or in combination. Truncation of the α3 N-terminus, in contrast to truncation of α1, had no effect on the release of Pba1-Pba2 (Fig. 5F). Truncation of the α3 N-terminus, however, abrogated the defect of Pba1-Pba2 release caused by the α1 truncation. These data indicate that the N-terminus of α1 has a critical role in maintaining the arrangement of the α-subunit N-termini in the assembling CP in a way that is compatible with Pba1-Pba2 release by keeping the α3 N-terminus in an open conformation as long as the Pba1 N-terminus is located in the α-ring pore.

## Discussion

Assembly of the eukaryotic proteasome is a complex process involving several assembly chaperones. To structurally characterize intrinsically short-lived assembly intermediates of CPs, we employed yeast cells that accumulated higher than WT levels of early PCs because of reduced efficiency of a downstream step, the β7-driven dimerisation of 15S-PCs. Utilizing this strategy allowed us to purify sufficient amounts of all intermediates that form by the incorporation of β1, β5, and β6

into the 13S-PC (consisting of α1-7, β2-β4, as well as Pba1-Pba2 and Ump1) to obtain six high-resolution cryo-EM structures (Fig. 1). Comparison of these structures with each other as well as with a late-PC resulting from β7-driven 15S-PC dimerisation[19], allowed us to dissect the order of events and many structural transitions that occur along with the different assembly steps.

Structural and genetic characterization of the early PCs suggests alternative assembly pathways for subunits β1, β5, and β6 into the 13S-PC. Although our 13S-PC and 13S + β5 + β6-PC are fully compatible with already published structures[13,21], we identified three previously undetected intermediate structures that indicate alternative assembly pathways between the 13S-PC and the 15S-PC. Specifically, we characterised 13S + β1- and 13S + β5-PCs, suggesting that either β1 or β5 can be the first subunits to join the 13S-PC (Fig. 6). Our biochemical study (Fig. 2B) as well as previous studies employing siRNA-mediated knockdown of individual proteasomal subunits in human cells, also suggest that an assembly route via β1 is possible[36]. In our cryo-EM dataset, 13S + β5-PCs are approximately three times more abundant than β1-associated 13S complexes, suggesting either a better availability of β5 or a higher stability of β5-containing complexes. Consistent with the existence of alternative assembly routes, we obtained all downstream intermediates of the two pathways (13S + β1 + β5-PC and 13S + β5 + β6-PC), indicating that β1 can enter the complex before or after β5 and/or β6. The assembly route taken from the 13S to 15S-PC might mostly be determined by the availability of β1, β5, and β6. Our characterisation of assembly intermediates in yeast indicates a hierarchy of subunit addition, wherein β5 is predominantly incorporated prior to the competitive or sequential recruitment of β1 and β6. This is in line with previous studies showing that β5 availability is rate-limiting in proteasome assembly[37].

The pivotal role of Ump1 in PC maturation is underscored by the observation that mutations or changes in Ump1/POMP are linked to human genetic disorders termed PRAID (POMP-related autoinflammation and immune dysregulation), and pathogen resistance in plants[38,39]. Structurally, Ump1 is crucial for the stepwise assembly of proteasomal PCs, as it provides contact surfaces for β-subunits, incorporation of which in turn promotes successive folding of the chaperone. The established structures of Ump1 and the β2 and β5 propeptides confine the space in the newly formed proteasomal antechamber. This confinement is further exacerbated when β7 is incorporated into the β-ring, pushing β6 and β1 into their final positions. Our work confirms findings for human proteasome biogenesis, where the S3/S4 loops remain flexible during β-ring assembly, at least until 15S-PC formation[21]. In our early PC structures, this is particularly evident in the case of β2, where the catalytic triad residues Thr[1], Asp[17], and Lys[33] are already positioned in hydrogen bonding distance, although the β2-Lys[33] side chain density is poorly resolved due to the flexibility of the S3/S4 loop (full-residue Q-score of 0.563 in the 15S-PC, σ = 0.6). In theory, proper positioning of Lys[33] allows for deprotonation of Thr[1] to initiate autocatalytic precursor processing. But for the reaction to proceed, Asp[166] needs to provide a proton via Ser[129] to Thr[1] [29]. Besides flexibility of the β2-Asp[166] side chain, β2-Ser[129] is still located roughly 5 Å away from the Thr[1] amino group, preventing autocatalytic activation of β2 in the 15S-PC (Fig. 3). We cannot identify any residues in the early PC structures that would sterically hinder the folding of the β2 S3/S4 loop. We therefore propose that the conformational freedom in these complexes is still too large to allow folding of the β2 S3/S4 loop. Even in the 13S-13S + β5 dimer, in which the conformational space of β2 is non-naturally restricted by the opposite α-ring, the β2 S3/S4 loops in both halves remain unfolded. We conclude that folding of the β2 S3/S4 β-hairpin loop is most likely induced only by proper 15S-PC dimerisation and interaction with helix H4 of the opposite β-ring, which likely also positions Lys[33] correctly to deprotonate the Thr[1] oxygen.

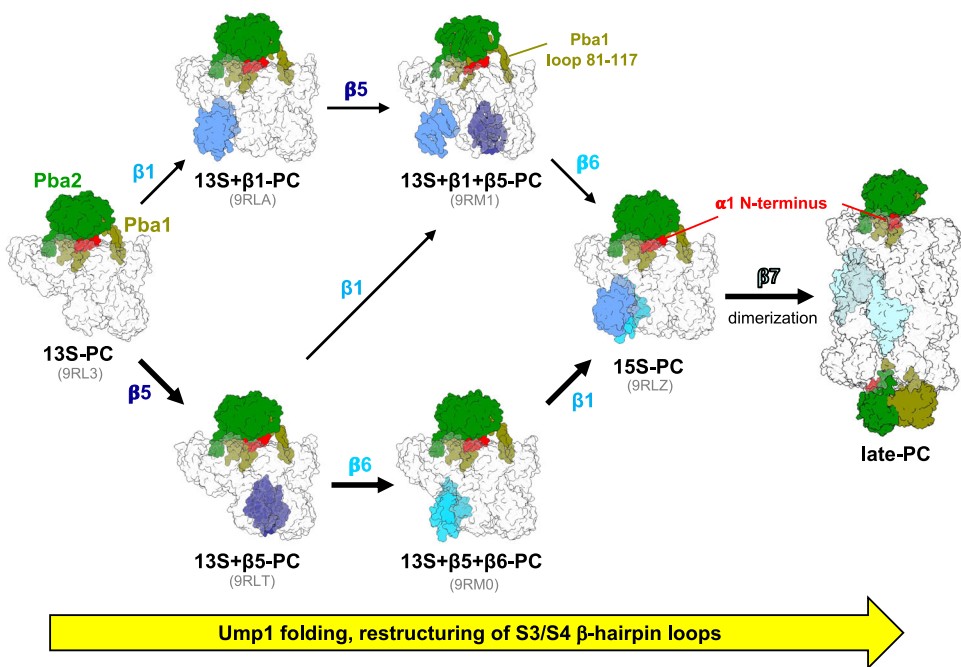

**Fig. 6 | Structure-based schematic model of late-PC assembly from the 13S-PC via 15S-PCs.** Shown is a summary of the main findings based upon cryo-EM structures of six 13S- to 15S-PCs isolated from *S. cerevisiae* (PDB codes in brackets). The representation of the late-PC was generated from PDB entry 8RVL. Incorporation of β1, β5, and β6 does not follow one fixed order, as β1 and β5 can assemble with 13S-PCs independently of each other. Higher abundance of complexes containing β5 and lacking β1, instead of vice versa, suggests that the bottom pathway is more often taken than the top one (as is indicated by the stronger arrows). Comparison of complexes revealed a structural transition relating to the Pba1-Pba2 chaperone, a loop of which (residues 81-117) is inserted into a pocket between subunits α3 and α4 in all analysed early PCs, but is not detectable in this position after β7 incorporation-triggered late-PC formation. The N-terminus of α1, the presence of which is of critical importance for the release of Pba1-Pba2 upon late-PC maturation, is highlighted in all shown PCs. The gradual folding of Ump1 and restructuring of S3/S4 hairpin loops, important for later autoactivation of nascent CPs, is indicated by the yellow arrow below.

In addition, our analysis of the early PC structures led to the identification of a previously unrecognized loop (residues 81-117) in the Pba1 chaperone. This loop is detected in all our early PC structures and intercalates between subunits α3 and α4, keeping α3 from reaching its final position in the ring (Fig. 5A). In the absence of α3 (*pre9Δ* background), the Pba1-Pba2 chaperone cannot bind to PCs, suggesting that contacts mediated by the identified loop initiate Pba1-Pba2 binding to early PCs. In addition, this loop is not resolved in the late-PC structure, indicating that its release from the α3/α4 pocket may go along with dimerisation of two 15S-PCs. We also identified a rearrangement of the α1 N-terminus (residues 3-16), which alters the interactions of Pba1-Pba2 with PCs during the transition from 15S-PC to late-PC. Notably, α1 N-terminal truncation leads to a defect in the release of Pba1-Pba2 from nascent CPs (Fig. 5F and Supplementary Fig. 12B), suggesting that this α1-Pba1 interaction is important for proper binding and release of the chaperone. The spatial proximity between the N-termini of α1 and α3 in the α-ring pore suggests that the α3 N-terminus takes the place of the α1-N-terminus when the latter is missing, thereby preventing the release of the Pba1 N-terminus from the pore. In support of this model, the concomitant deletion of the α3 N-terminal residues rescued the defect in Pba1-Pba2 release caused by the α1 truncation. Hence, our work suggests that release and recycling of the Pba1-Pba2 chaperone follows a stepwise sequence of events. These events coordinate PC maturation and the ordered stacking of the α-subunits N-termini with the release of binding sites for 19S RP or other activators.

Altogether, our study highlights the alternative assembly pathways between 13S and 15S complexes in yeast, the gradual folding of Ump1, the restructuring of β-subunit S3/S4 loops, and the importance of the α1 N-terminus for Pba1-Pba2 release, which is associated with loosening of the so far uncharacterised loop in Pba1 (residues 81-117)

intercalated between α3 and α4, summarized in Fig. 6 and Supplementary Movie 1.

## Methods

### Protein expression, purification and analysis

Yeast strains used for purification and biochemical analyses of proteasomal precursor complexes (PCs) are listed in Table S6. Strain MO27 (*pre4-ΔC19 FLAG-6xHis-UMP1 blm10-Δ P_GAL1-PBA1 P_GAL1-PBA2*) was used for purification of early PCs, taking advantage of the 6xHis- and FLAG-tags[20]. MO27 cells were grown in 4 liter complete medium supplemented with galactose as a carbon source. The N-terminal FLAG-6His tag on Ump1 allowed purification of Ump1-containing PCs. Overexpression of Pba1-Pba2 and deletion of the *BLM10* gene were important to increase the yield of Pba1-Pba2-capped complexes. A tandem affinity purification protocol taking advantage of the 6His and FLAG tags was performed as follows. Cells were pelleted and frozen in liquid nitrogen. The cells were ground to powder using a Mixer Mill MM400 (Retsch) for 1 min at 30 Hz. Proteins were extracted by incubating the powder with FLAG buffer (50 mM Tris pH 7.5, 5 mM MgCl2, 2 mM ATP, 150 mM NaCl, and 15% (v/v) glycerol) supplemented with 10 mM imidazole and protease inhibitors (cOmplete, EDTA-free, Cat. No. 11836170001, Roche Diagnostics), in the proportion 2 ml per g wet weight, for 30 min at 4 °C with rotation. Cell debris was removed by centrifugation at 30,000xg for 30 min at 4 °C, and the supernatant was then passed through a 0.2 μm filter. The filtrate was incubated with Ni-NTA Superflow resin (Cat. No. 30430, Qiagen) equilibrated in FLAG buffer for 1 hour at 4 °C with agitation. The resin was washed 4 times with 10 column bed volumes of FLAG buffer supplemented with 10 mM imidazole. Bound complexes were eluted in two steps with FLAG buffer supplemented with 50 mM and 100 mM imidazole, respectively. Eluted fractions were diluted 2.5 times in FLAG buffer and further incubated with EZview Red ANTI-FLAG M2 Affinity Gel (Cat.No. F2426-

1ML, Batch SLCL4336, Sigma-Aldrich) for 90 min at 4 °C with rotation. The beads were washed 3 times with 20 column volumes of FLAG buffer. Bound proteins were eluted three times (E1-E3) with one column volume each of FLAG buffer without ATP and without Glycerol containing 0.3 mg/ml FLAG peptide (Cat. No. F3290, Sigma-Aldrich). Eluted material was then concentrated using Vivaspin 6 columns 50,000 MWCO (Cat. No. GE28-92-2318, Sigma-Aldrich). Protein concentration of the eluted material was about 0.4 mg/ml. MO24 (*PRE1-FLAG-6His*) was used to purify wild-type 20S CPs as a control in native PAGE analysis using the same two-step procedure, except that cells were grown in 1 liter of complete medium with glucose (YPD). Protein concentration of the purified material was about 0.4 mg/ml. Samples were diluted with cold sample buffer (240 mM Tris-HCl, pH 8.8, 0.04% (w/v) bromophenol). Purified complexes were analysed under native conditions in 4-16% Bis-Tris Gels (Cat. No. BN1002Box, Invitrogen), for which the samples were diluted with cold sample buffer (240 mM Tris-HCl, pH 8.8, 0.04% (w/v) bromophenol). Electrophoresis was performed on ice at 16 mA/gel for around 1.5 hours. The proteins in the gel were visualized by silver staining.

Constructs (pJD844/845 and pJD847) for the generation of α1/*scl1* and α3/*pre7* (pJD847) mutant yeast strains, respectively, were generated by overlap extension PCR to delete internal parts of the *SCL1* or *PRE7* ORFs (primers used are listed in Table S7). Two variants were created for α1/Scl1, one deleting amino acid residues 2-16 (α1ΔNT(MT)), and another replacing these residues with a glycine residue (α1ΔNT(MGT)); for α3/*pre7*, one version deleting residues 2-10 was generated (α3ΔNT). The PCR products were cloned into the integrative yeast vector YIplac211[40]. Using a two-step gene transplacement strategy, the mutant versions were then introduced stably into the genome of yeast strains expressing 2xHA-tagged versions of Ump1 or Pba2[41]. The resulting yeast mutant strains (Supplementary Table 6) were grown to an $OD_{600nm} = 1$ in 25 mL cultures. Cells were washed in $H_2O$ and resuspended in 75 μL lysis buffer (50 mM Tris pH 7.5, 5 mM $MgCl_2$, 2 mM ATP, 1 mM DTT, 15% (v/v) glycerol and 1x protease inhibitor (cOmplete, EDTA-free, Cat. No. 11836170001, Roche Diagnostics)). Crude lysates were generated by vortexing in the presence of glass beads. After a clearing spin (15 min 30,000 x *g* at 4 °C), 8 μg of total protein were analysed by native PAGE in 3–12% Bis-Tris gel (Cat.No. BN1001BOX, Invitrogen), western blotting onto PVDF membranes (Cat.No. 03010040001, Roche), and detection using peroxidase-coupled rat monoclonal (clone 3F10) anti-HA antibody (Cat.No. 12013819001, Merck/Sigma-Aldrich, at 1:2500 dilution) and enhanced chemiluminescence (ECL) reagent SuperSignal West Femto (Cat.No. 34095, Thermofisher) and Xray film.

To investigate the order of β-subunit assembly in vivo, we generated a set of strains in which the β1 subunits carry an N-terminal FLAG-tag (using a two-step gene transplacement strategy and vector YIplac211[40]). For this purpose, a DNA segment was created by overlap extension PCR that encodes FLAG-tagged Pre3/β1, and cloned into YIplac211[40] after cleavage with SacI and HindIII (Supplementary Table 7). FLAG-tagged Pre3 was introduced either into strains, in which the β5-encoding gene *PRE2* or the β6-encoding gene *PRE7* were placed under the control of the glucose-repressible $P_{GAL1}$ promoter using a one-step gene transplacement strategy and the oligos listed in Supplementary Table 7[28,42]. FLAG-pro-β1-containing complexes were collected and analysed as follows. Cells were grown in 50 mL minimal media containing galactose till OD 0.5 to 0.6 and then collected, resuspended in 200 mL minimal media containing glucose, and incubated further for 15 hours. Cells were harvested, frozen, and kept at 80 °C. Pelleted cells were ground into powder in a mortar in the presence of liquid nitrogen. Tris buffer (50 mM, pH7.4) supplemented with 150 mM NaCl and protease inhibitors was then added in the ratio 1 mL per gram fresh cells. Cell debris was removed by centrifugation at 30,000 x *g* for 30 min at 4 °C. 400 μl crude extract were incubated with 50 μl EZview red anti-FLAG M2 affinity gel (see above) for 1 hour at 4 °C

with mild shaking. Flow-through was removed and the resin was washed twice with 40 column volumes of extraction buffer. Bound complexes were eluted in 100 μl extraction buffer containing 300 μg/ml FLAG peptide (see above). A strain without a FLAG tag served as a control. Total extract proteins (input control) and eluted complexes were separated by native PAGE in 3-12% Bis-Tris gels (see above). Protein complexes were transferred to a 0,2 μm Amersham nitrocellulose membrane (Cat.No. 10600001, Avantor/VWR) during 2 hours at 0.8 mA/cm² in a semidry blot transfer chamber. Membranes were incubated overnight with rabbit polyclonal antibodies anti-Prs1 (1:1000)[27] and rat monoclonal (clone 3F10) anti-HA antibody (Cat.No. 11867431001, Lot-No. 62572200, Merck/Sigma-Aldrich, at 1:2500 dilution). Membranes were washed 4x with PBS and incubated for one hour with the secondary goat anti-Rabbit, Alexa Fluor Plus 800-coupled IgG (Cat.No. A32730, Lot-No. VL313935, Thermo Fisher Scientific) and the goat anti-Rat, Alexa Fluor 680-coupled secondary IgG (Cat. No.A-21096; Lot-No. 2409044, Thermo Fisher Scientific). Signals were scanned with the Li-Cor Odyssey Infrared Imager and processed with Image Studio version 5.2.

## Electron microscopy sample preparation and data acquisition

Before plunge freezing, 6 μl of the concentrated flag elution (-8.2 mg/mL) was supplemented with 2 μl of 0.4% CHAPSO. Immediately afterwards, 5 μl of the resulting solution was applied to a freshly glow-discharged Quantifoil UltrAuFoil grid and incubated for 45 s at 4 °C and 85% humidity. Excess liquid was manually blotted away for 11 s, and the grid was plunge frozen in liquid ethane using a manual plunge freezing device (Neptune fluid flow systems). Screening of cryo-EM conditions was done on a Talos F200C transmission electron microscope. High-resolution cryo-EM data were collected on a Titan Krios G3i transmission electron microscope (Thermo Fisher Scientific) operated at 300 kV equipped with a BioQuantum GIF post-column energy filter (Gatan) and a K3 direct electron detector (Gatan). 28,764 micrographs were recorded in counting mode at a pixel size of 0.834 Å. The defocus range was set between −0.8 and −2.8 μm. Each micrograph was dose-fractionated to 37 frames with a total exposure time of 1 s and a total electron dose of roughly 65 e⁻/Å². Automated data collection was done using EPU (Thermo Fisher Scientific).

## Cryo-EM image analysis

EM-data processing was done using cryoSPARC version 4.5.3[43]. Movie frame alignment and dose weighting were performed with Motion-Cor2, and contrast transfer functions were determined via CTFFIND4[44,45]. All refinements used gold-standard Fourier shell correlation (FSC) calculations, and reported resolutions are based on the FSC = 0.143 criterion of mask-corrected FSC curves. All refinement jobs were done without applying any symmetry. The processing strategy is visualized in Supplementary Fig. 2. In short, low-quality micrographs with an estimated resolution of over 5 Å, an estimated defocus of -3.0 μm, or a relative ice thickness over 2.0 were discarded. Particle picking was done template-based using 2D templates generated from particles picked via cryoSPARCs blob-picker. Low-quality particles were then removed through three iterative rounds of 2D classification, followed by subset selection. Afterwards, three ab initio models were generated, and the particles were assigned to these models via heterogeneous refinement. The particle stack corresponding to the 13S-13S + β5 dimer was further cleaned by 3D classification and reconstructed using non-uniform refinement in cryoSPARC. Following heterogeneous refinement of the initial particle stack, the isolated early PCs and the 13S-15S dimer were separated by one round of 2D classification and subset selection. Particles representing the 13S-15S dimer were used for ab initio reconstruction, followed by 3D reconstruction via non-uniform refinement. The early PCs were separated by masked 3D classification and refined individually using masked non-uniform refinement. Particle numbers, applied symmetry settings,

and global resolutions of the reconstructed complexes are summarized in Table 1 and Supplementary Fig. 2.

## Molecular modelling

All pdb-models were built using relevant chains (α1-7, β1-6, Ump1, Pba1 and Pba2) from PDB entry 8RVL[19] as a starting model. The resulting initial pdb-models were fitted into the electron densities using UCSF ChimeraX[46]. The N-terminal FLAG-6xHis tag attached on Ump1 was manually build using COOT[47] (Supplementary Fig. 13). The final pdb-models were built in iterative cycles of manual building with COOT and real-space refinement in PHENIX[48,49]. The data collection and model statistics are summarised in Table 1.

## Data visualisation

Structure visualization and comparison was done using UCSF ChimeraX[46]. All contacts are calculated using UCSF ChimeraX with VDW overlap ≥ -0.4 Å, ignoring interactions between residues <2 apart in sequence. Hydrogen bonding distances and angle criteria are relaxed by initial default settings (distance tolerance 0.4 Å, angle tolerance 20°) as recommended by ChimeraX.

## Reporting summary

Further information on research design is available in the Nature Portfolio Reporting Summary linked to this article.

## Data availability

The cryo-EM maps have been deposited in the Electron Microscopy Data Bank (EMDB) under accession codes, EMD-54029 (13S-PC), EMD-54032 (13S + β1-PC), EMD-54048 (13S + β1 + β5-PC), EMD-54047 (13S + β5 + β6-PC), EMD-54046 (15S-PC) and EMD-54045 (13S-13S + β5-dimer), respectively. Atomic coordinates and structure factors derived from the EM maps have been deposited in the Protein Data Bank under accession codes 9RL3 (13S-PC), 9RLA (13S + β1-PC), 9RM1 (13S + β1 + β5-PC), 9RM0 (13S + β5 + β6-PC), 9RLZ (15S-PC), and 9RLT (13S-13S-β5-dimer). Previously published models used for comparisons can be found at: 7LSX, 7LS6, 8RVL, 8RVO, 8RVP, 8RVQ, 5CZ4, 6EF3, 7TEJ, 8TM5, 8QYM, 8QZ9, 8QYN, and 8YIX. Source data are provided with this paper.

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

## Acknowledgements

This work was funded by the Deutsche Forschungsgemeinschaft (DFG, German Research Foundation) project numbers 406260942 (P. W.) and 442219341 (P.W. and J. D). We acknowledge access to electron microscopic equipment at the Core Facility for cryo-Electron Microscopy (CFcryoEM) of the Charité - Universitätsmedizin Berlin supported by DFG (INST 335/588-1 FUGG) for cryo-EM data collection, and we thank Dr. Thiemo Sprink and Dr. Christoph Diebolder for data collection. The authors would like to thank Kerstin Nürrenberg, Fleur Kayser, and Melanie Anding for technical assistance and Dr. Maximilian Voit for IT support.

## Author contributions

M.M.N., A.C.M., P.C.R., and R.J.D. designed and performed the genetic and biochemical experiments. P.C.R and E.M. purified all protein complexes for biochemical analyses and EM sample preparation. P.W. and E.M. designed the EM studies. The EM-specimen preparation and data processing were done by E.M. EM-data interpretation was carried out by E.M. and P.W. P.W., E.M., P.C.R,. and R.J.D. wrote and edited the manuscript. Project funds were acquired by P.W. and R.J.D.

## Funding
