## [Transparent Peer Review file · Nature Communications]

Structural transitions in the stepwise assembly of proteasome core particles

Corresponding Author: Professor Petra Wendler

Version 0:

Reviewer comments:

Reviewer #1

(Remarks to the Author)

In this manuscript by Mark et al, several structures of proteasomal core particle subcomplexes are reported. Some of these complexes are known assembly intermediates, and some are self-reported by the authors to be purification/concentration artifacts or previously unresolved assembly products that represent alternative assembly pathways. Based on analyses of these structures and a single biochemical experiment, the authors propose: i) multiple beta subunit assembly sequences exist in yeast and possibly humans; ii) that S3/S4 beta hairpin loops serve both to stabilize incoming beta subunits and to orient residues in the catalytic triad for propeptide removal; and iii) a network of interactions that, when disrupted, result in the release of Pba1-2 from the maturing CP.

Overall, the structures are a helpful complement to a large number of structures already published containing various CP intermediates, and they reveal what are likely additional regulatory mechanisms that, if true, would expand our understanding of CP biogenesis. The manuscript is clearly written, and the figures are generally clear. However, there are some concerns that the mechanisms proposed rely entirely on the structures, and are not validated via any secondary mechanisms (save for a sole biochemical experiment). Thus, the manuscript would be strengthened significantly if the evidence for the mechanisms proposed weren't based entirely on structures.

Major comments:

1. As mentioned above, the major limitation of the manuscript is that, with the exception of the novel interaction proposed for the Alpha1 N-terminus, the authors largely assume that the functions and events inferred from their structures occur as they state, without explicit validation (e.g., via mutagenesis or other experiments).

A good example of this is that, even in the abstract, the authors explicitly claim that structures harboring distinct, overlapping subsets of beta subunits is "showing that alternative assembly pathways to the 15S-PC exist in yeast," rather than considering the possibility that they are partially disassembled species or off-pathway assembly products that resulted from depletion of the catalytic amounts of Pba1-2 that occurs in response to Beta7 mutation (the authors state this occurs on pg. 4, lines 131-133). Although the authors cite some circumstantial evidence from previous work that is consistent with their observations, those studies also required potentially damaging genetic perturbations or substantial manipulation of precursors that could have induced subunit dissociation.

Provision of some direct evidence that these are likely to be assembly intermediates rather than dissociation products is required to substantiate this claim. As an example for the alternative assembly pathways, can the authors demonstrate that Beta5/6 dissociates from the 15S-PC more readily than Beta1 upon increasingly harsh buffer conditions as would be expected for such a pathway? Can the authors directly demonstrate addition of Beta5 and/or Beta6 to 13S-PCs containing Beta1, or demonstrate that Beta1 directly adds to the 13S-PC containing only Beta2-4?

2. The repositioning of the N-terminus of Alpha1 to promote Pba1/2 dissociation is intriguing, but the mode of transmission proposed by the authors (Alpha1-

Ser8 Alpha3-Arg4/Arg5 Alpha4-Arg6 Pba1-Glu11) is unlikely, given that proteasomes lacking the N-terminal 10 residues of Alpha3 appear to assemble and mature normally. As more meaningful support for this model, disruption of the residues in this proposed cascade aside from solely Scl1/Alpha1 should be assayed for an effect on Pba1/2 similar to that shown for Alpha1.

Minor comments:

1. In Figure 5F, a loading control showing that Pba2-HA is expressed in the (otherwise) WT strain should be included.
2. Pg. 2, line 73 the authors used "und" instead of "and."

3. Pg. 3., line 109: "b-subunits" should be "β-subunits" to match the rest of the manuscript.
4. Pg. 5, line 200: reference #21 appears to be in a different format than the rest of the manuscript.
5. Pg. 5, lines 210 and 211: there appears to be a space missing between the word residues and the residue numbers.
6. Pg. 8, line 336: hypothesize is misspelled.

Reviewer #2

(Remarks to the Author)

The manuscript by Mark et al. reports on the cryo-EM structure analysis of the yeast 15S-PC and several intermediates reflecting stepwise assembly of beta-subunit to the CP precursor complex 13S-PC. I found the work is structurally interesting to the proteasome field and bridges an important missing gap in understanding the biogenesis pathway of the proteasome core particle. Nonetheless, there are a number of structural studies published in the past several years that have made notable progress toward understanding the structural basis of the proteasome biogenesis, which overshadows the present manuscript and makes it less appealing and more like an incremental advance. In addition, there remains a certain level of uncertainty or potential risk of over-interpretation in the present form, given the lack of additional biochemical tests. I think this paper is a potentially good candidate for publication in Nature Communications, if the authors can address the following issues in a revision.

(1) While the paper provides ample structural models, there is no data to either validate or support that the observed 13S to 15S-PC intermediate complexes are indeed on the pathway of proteasome biogenesis. Instead, the authors took it for granted that they have observed "all intermediates resulting from step wise addition of beta-subunits ...", without any additional evidence from complementary methods like functional analysis, mutagenesis, and/or mass spectrometry etc. What if these complexes were from partially disassembled intermediate of 15S-PC in a poorly controlled, disfavored condition, therefore representing off-pathway intermediates? What if these complexes can only survive in the in vitro condition, but non-existent in vivo? Believing that they are the stepwise assembly intermediate is foremost a hypothesis or speculation if no other evidence can help consolidate their true functional identity. If the authors cannot fully justify their conclusion, they should instead turn down the tone and address necessary caveats and limitations of this study.

(2) Another way to help justify the hypothesis of claiming that the solved structures are on-pathway assembly intermediates is to make thorough structural comparison of these structures with other CP assembly intermediates previously published, whenever feasible. These comparative studies should be presented as part of main figures or supplementary figures to support the description in pages 4-5 and will at least help verify which part of the assembly is rationally expected or temporally compatible with the existent structural models in the literature. If there are apparent incompatible features in the comparison, the authors should seek complementary experiment to make necessary reconciliation.

(3) The local EM densities of S3/S4 beta-hairpin loops and Pba1 loop that inserts into the alpha pocket must be presented in either main figures or supplementary figures to support the authors' claim that they indeed solved these loops to high enough resolution that allowed their accurate atomic modeling and conclusion. Similarly, the local EM densities of the N-terminal of alpha1 must be presented for visual assessment to support their atomic and functional interpretation. More generally, any "previously unrecognized details" must be supported or justified by clear presentations of high-resolution EM densities in either main figures or supplementary figures.

(4) In Table S1, why is the percentage of disallowed residues all above 90%? This is either done wrong, or the table contains typo.

(5) The introduction could be improved in the clarification of what CP assembly intermediates have been structurally solved, what is yet to be solved, and why the present study is important. The discussion section may also be improved with a summary on the key differences between the newly solved structures and the previously published CP assembly intermediate structures.

(6) It would be very helpful to add a new main figure to summarize the main findings on the assembly mechanism. The logic flow of the paper is a little bit brittle and lacks focus of a central story. This may be improved with better mechanistic discussion and conclusion paragraphs at the end of the paper.

Minor issues:

(1) Please present gel filtration or size-exclusion chromatographic results of the purified complexes in supplementary figures.

(2) There are numerous places (e.g. line 70, 73, 74, 109, 114, etc.) where Greek alpha is written as "a", and beta is written as "b".

(3) Table S1 may have to be moved to main text as a main Table.

Version 1:

Reviewer comments:

Reviewer #1

(Remarks to the Author)

In this revised manuscript, Mark et al. included additional data to address a number of concerns raised in the initial round of review, with the foremost being that the Beta1-containing 13S precursor complex was not convincingly shown to be an "on-pathway" assembly product, rather than an artifactual dissociation product or an artifact of limiting amounts of assembly chaperones that might result in altered subunit compositions. The authors have addressed this by overproducing Pba1-2 to prevent their depletion, and have demonstrated the ability to detect Beta1 in yeast cells expressing vanishingly low levels of

Beta5 or Beta6, which have traditionally been considered to incorporate prior to Beta1. Although this data does not unequivocally prove a Beta1-first assembly mechanism, it does help to further substantiate this possibility and reduce the likelihood of alternative explanations. Text changes to the abstract and main article have softened this claim to a level this Reviewer believes is appropriate.

Some minor edits are noted below:

1. pg. 2, line 57: the term ortholog might be a better word choice than alias for describing the Ump1 ortholog in humans.
2. Pg. 4, line 148: the word "too" should be "to".

Reviewer #2

(Remarks to the Author)

The revised manuscript has sufficiently addressed all my questions. I recommend its publication in the present form.

Point-by-Point Response to Reviewer Comments

We sincerely thank both reviewers for their insightful, constructive and appreciative feedback, which has served as a valuable catalyst for additional experiments and significant revisions to our manuscript. We believe that the resulting improvements have substantially strengthened the clarity and overall quality of the manuscript.

Reviewer #1

In this manuscript by Mark et al, several structures of proteasomal core particle subcomplexes are reported. Some of these complexes are known assembly intermediates, and some are self-reported by the authors to be purification/concentration artifacts or previously unresolved assembly products that represent alternative assembly pathways. Based on analyses of these structures and a single biochemical experiment, the authors propose: i) multiple beta subunit assembly sequences exist in yeast and possibly humans; ii) that S3/S4 beta hairpin loops serve both to stabilize incoming beta subunits and to orient residues in the catalytic triad for propeptide removal; and iii) a network of interactions that, when disrupted, result in the release of Pba1-2 from the maturing CP.

Overall, the structures are a helpful complement to a large number of structures already published containing various CP intermediates, and they reveal what are likely additional regulatory mechanisms that, if true, would expand our understanding of CP biogenesis. The manuscript is clearly written, and the figures are generally clear. However, there are some concerns that the mechanisms proposed rely entirely on the structures, and are not validated via any secondary mechanisms (save for a sole biochemical experiment). Thus, the manuscript would be strengthened significantly if the evidence for the mechanisms proposed weren't based entirely on structures.

Major comments:

1. As mentioned above, the major limitation of the manuscript is that, with the exception of the novel interaction proposed for the Alpha1 N-terminus, the authors largely assume that the functions and events inferred from their structures occur as they state, without explicit validation (e.g., via mutagenesis or other experiments).

A good example of this is that, even in the abstract, the authors explicitly claim that structures harboring distinct, overlapping subsets of beta subunits is "showing that alternative assembly pathways to the 15S-PC exist in yeast," rather than considering the possibility that they are partially disassembled species or off-pathway assembly products that resulted from depletion of the catalytic amounts of Pba1-2 that occurs in response to Beta7 mutation (the authors state this occurs on pg. 4, lines 131-133). Although the authors cite some circumstantial evidence from previous work that is consistent with their observations, those studies also required potentially damaging genetic perturbations or substantial manipulation of precursors that could have induced subunit dissociation.

Provision of some direct evidence that these are likely to be assembly intermediates rather than dissociation products is required to substantiate this claim. As an example for the alternative assembly pathways, can the authors demonstrate that Beta5/6 dissociates from the 15S-PC more readily than Beta1 upon increasingly harsh buffer conditions as would be expected for such a pathway? Can the authors directly demonstrate addition of Beta5 and/or Beta6 to 13S-PCs containing Beta1, or demonstrate that Beta1 directly adds to the 13S-PC containing only Beta2-4?

Our Response: We thank the reviewer for the overall positive evaluation of our work. We acknowledge the reviewer's reasonable concern that some of the precursor complexes structurally characterized in this study may not constitute physiologically relevant precursors in the biogenesis pathway leading to the mature 20S proteasome. Because it is technically next to impossible to biochemically follow the 20S assembly in real time *in vivo* in wild-type cells, we have opted to accumulate 13S-15S precursors by slowing down downstream dimerisation of 15S PCs using a strain lacking C-terminal residues of the $\beta 7$ subunit. We chose this mutation because the $\beta 7$ subunit itself is not part of these complexes. The impaired 15S-PC dimerization kinetics resulting from this mutation lead to a depletion of the Pba1-Pba2 chaperone due to a reduced efficiency of its recycling, which occurs only upon final maturation of the proteasome. This problem was addressed by the simultaneous overexpression of both subunits of this chaperone. As intended, our isolated pool of proteasomal precursors is devoid of precursor complexes lacking Pba1-Pba2. To our knowledge, our study is unique in isolating native yeast 13S-15S complexes bearing no mutations and with sufficient chaperones present to prevent off-pathway assemblies. In principle, however, a general limitation of purification protocols is the inherent risk of disrupting fragile protein complexes. As the reviewer posits, the emergence of the unique 13S+ $\beta 1$ -PC species from later-stage PC assemblies might occur by a non-physiological dissociation of the $\beta 5$ or $\beta 5/\beta 6$ subunits from those complexes. The isolated proteasomal pool contains as much 13S+ $\beta 5$ -PC as all subsequent PCs combined, while 13S+ $\beta 1$ -PC is underrepresented. We

therefore suspect that $\beta 5$ binds better to PC or is more readily available than $\beta 1$. Hence a scenario where $\beta 5$ or $\beta 5/\beta 6$ subunits dissociate non-physiologically seems unlikely, as it would require the selective dissociation of $\beta 5$, but not $\beta 1$ from only a small subset of the PC intermediates. Still, since both reviewers expressed similar concerns about our conclusions regarding the alternative assembly pathway starting with $\beta 1$ addition to 13S-PCs, we addressed these concerns experimentally using an *in vivo* approach. Specifically, we created a strain that allowed to selectively shut down expression of the $\beta 5$ subunit in a strain with a $\beta 1$ subunits carrying an N-terminal FLAG tag appended to its propeptide. To verify efficient shut down of $\beta 5$, this subunit carried a C-terminal 2xHA tag to enable detection of the subunit with a high sensitivity. The N-terminal FLAG tag on $\beta 1$ subunits allowed us to pull down PCs that had incorporated this subunit. As shown in the new figure 2B, in contrast to an untagged control, FLAG-tagged $\beta 1$ pulled down 13S-PCs in the absence of either $\beta 5$ or $\beta 6$. These findings clearly establish that $\beta 1$ has the propensity to assemble with 13S-PCs without prior addition of $\beta 5$ and/or $\beta 6$ and therefore suggest that the observed complexes containing $\beta 1$ but not $\beta 5$ and/or $\beta 6$ are likely assembly products and not the result of complexes resulting from dissociation of $\beta 5$ and/or $\beta 6$. These *in vivo* data are therefore in line with our conclusion based upon our isolation and structural characterization of 13S+ $\beta 1$ -PCs, and thus provide further support for an alternative assembly pathway, in which $\beta 1$ is assembling with 13S-PCs before the addition of $\beta 5$ and $\beta 6$. The observation that all intermediates of this alternative pathway were identified in our structural analysis (and supported by the above-mentioned genetic approach), furthermore suggests that these complexes are not results of non-productive off-pathways. While the above-mentioned biochemical and structural evidence supports the feasibility of an alternative assembly route, the observed relatively larger amounts of 13S+ $\beta 5$ -PC compared to 13S+ $\beta 1$ -PC suggest that $\beta 5$ more often is the first subunit to assemble with 13S-PCs. We have adjusted our wording of this issue in the abstract, main text and discussion, and adjusted the figure 2A to highlight (by a stronger arrow) that this pathway has a higher prevalence than the alternative pathway, in which $\beta 1$ appears to be the first subunit to join the 13S-PC.

2. The repositioning of the N-terminus of Alpha1 to promote Pba1/2 dissociation is intriguing, but the mode of transmission proposed by the authors (Alpha1-Ser8 \rightarrow Alpha3-Arg4/Arg5 \rightarrow Alpha4-Arg6 \rightarrow Pba1-Glu11) is unlikely, given that proteasomes lacking the N-terminal 10 residues of Alpha3 appear to assemble and mature normally. As more meaningful support for this model, disruption of the residues in this proposed cascade aside from solely Scl1/Alpha1 should be assayed for an effect on Pba1/2 similar to that shown for Alpha1.

Our Response: We thank the reviewer for the critical feedback on this part of our study, which prompted us to perform additional experiments to scrutinize and further develop our mechanistic explanation for the results obtained with N-terminally truncated $\alpha 1$. Our structural analysis suggests that the N-terminus of $\alpha 1$, which is in bonding distance to the loop segment aa121- 138 of Pba2 and the N-termini of $\alpha 2$ and $\alpha 3$, prevents the latter from prematurely closing the central pore in the α -ring such that the Pba1-N-terminus is trapped. To test this interpretation, we have created and analyzed a set of strains, in which the N-termini of $\alpha 1$ and $\alpha 3$ were truncated either alone, or in combination. Consistent with the reviewer's notion, truncation of the $\alpha 3$ N-terminus, in contrast to truncation of $\alpha 1$, had no effect on the release of Pba1-Pba2. Strikingly, truncation of the $\alpha 3$ N-terminus abrogated the defect of Pba1-Pba2 release caused by the $\alpha 1$ truncation. These data, shown in the new figure 5F, support our interpretation that the N-terminus of $\alpha 1$ has a critical role in maintaining the arrangement of α -subunit N-termini in the assembling core particle in a way compatible with Pba1-Pba2 release by keeping the $\alpha 3$ N-terminus in an open conformation as long as the Pba1 N-terminus is located in the α -ring pore. Please also see our response to reviewer 2 (1).

Minor comments:

1. In Figure 5F, a loading control showing that Pba2-HA is expressed in the (otherwise) WT strain should be included.

Our Response: The new figure 5F includes a lane showing that Pba2-HA is only detectable in 13S-15S precursors in wild-type cells, but not in late-PCs as is observed in strains with N-terminally truncated $\alpha 1$.

2. Pg. 2, line 73 the authors used "und" instead of "and."

3. Pg. 3., line 109: "b-subunits" should be " β -subunits" to match the rest of the manuscript.

4. Pg. 5, line 200: reference #21 appears to be in a different format than the rest of the manuscript.

5. Pg. 5, lines 210 and 211: there appears to be a space missing between the word residues and the residue numbers.

6. Pg. 8, line 336: hypothesize is misspelled.

Our Response: Thank you for pointing out these typos and formatting issues, which we now have fixed.

Reviewer #2:

The manuscript by Mark et al. reports on the cryo-EM structure analysis of the yeast 15S-PC and several intermediates reflecting stepwise assembly of beta-subunit to the CP precursor complex 13S-PC. I found the work is structurally interesting to the proteasome field and bridges an important missing gap in understanding the biogenesis pathway of the proteasome core particle. Nonetheless, there are a number of structural studies published in the past several years that have made notable progress toward understanding the structural basis of the proteasome biogenesis, which overshadows the present manuscript and makes it less appealing and more like an incremental advance. In addition, there remains a certain level of uncertainty or potential risk of over-interpretation in the present form, given the lack of additional biochemical tests. I think this paper is a potentially good candidate for publication in Nature Communications, if the authors can address the following issues in a revision.

(1) While the paper provides ample structural models, there is no data to either validate or support that the observed 13S to 15S-PC intermediate complexes are indeed on the pathway of proteasome biogenesis. Instead, the authors took it for granted that they have observed “all intermediates resulting from step wise addition of beta-subunits ...”, without any additional evidence from complementary methods like functional analysis, mutagenesis, and/or mass spectrometry etc. What if these complexes were from partially disassembled intermediate of 15S-PC in a poorly controlled, disfavored condition, therefore representing off-pathway intermediates? What if these complexes can only survive in the *in vitro* condition, but non-existent *in vivo*? Believing that they are the stepwise assembly intermediate is foremost a hypothesis or speculation if no other evidence can help consolidate their true functional identity. If the authors cannot fully justify their conclusion, they should instead turn down the tone and address necessary caveats and limitations of this study.

Our Response: We thank the reviewer for considering our work as structurally interesting to the proteasome field and bridging an important missing gap in understanding the biogenesis pathway of the proteasome core particle. We acknowledge the reviewer's reasonable concern regarding the lack of biochemical validation of our interpretations and would like to refer to our answer to major comment 1 of reviewer 1. As mentioned above, our new biochemical data show that complexes that contain $\beta 1$ but lack $\beta 5$ and/or $\beta 6$ are not just the result of *in vitro* disassembly of later intermediates, but can form *in vivo* in cells lacking either $\beta 5$ or $\beta 6$ due to an expression shutdown (new Figures 2B and Extended figure 3). These findings therefore underscore the ability of $\beta 1$ to assemble with 13S-PCs early on, before $\beta 5$ and $\beta 6$ have been incorporated. As discussed in some detail in the response to reviewer 1, while there is evidence that an early incorporation of $\beta 5$ is likely to be the prominent route of assembly, our findings suggest that there is some flexibility in the system.

As we tried to better highlight in the revised manuscript, we strongly believe that our finding, aside from providing more complete and higher resolution structures of 13-15S PCs than have been previously available (at least for yeast PCs) indeed bridge a number of important gaps in our understanding of CP assembly. Specifically, by structurally resolving four intermediates on the way from yeast 13S to the 15S-PC (half-CP without $\beta 7$), that have not been available before (yeast 13S+ $\beta 5$ -PC, 13S+ $\beta 1$ -PC, 13S+ $\beta 1$ + $\beta 5$ -PC, 15S-PC) we were able to monitor how multiple interactions of the Ump1 chaperone with incoming β -subunits not only contribute to their incorporation into the nascent complex, but also how these interactions help to gradually shape Ump1's final donut-like structure as it is highlighted in Figure 4A. Our structural resolution of these intermediates, which are free of mutations, allow to follow the stepwise restructuring of β -hairpin loops that prepare the complexes for the later CP activation by autocatalytic processing of β -subunit propeptides. We identify a previously overlooked loop of Pba1 that intercalates between subunits $\alpha 3$ and $\alpha 4$. Furthermore, our structures of early assembly intermediates point to a role of the $\alpha 1$ N-terminus in signalling CP maturation to Pba1-Pba2 release.

(2) Another way to help justify the hypothesis of claiming that the solved structures are on-pathway assembly intermediates is to make thorough structural comparison of these structures with other CP assembly intermediates previously published, whenever feasible. These comparative studies should be presented as part of main figures or supplementary figures to support the description in pages 4-5 and will at least help verify which part of the assembly is rationally expected or temporally compatible with the existent structural models in the literature. If there are apparent incompatible features in the comparison, the authors should seek complementary experiment to make necessary reconciliation.

Our Response: We have now added a new supplemental table S1 to the manuscript summarizing already published structures of β -ring assembly intermediates found in human and yeast cells (Schnell et al., 2021; Adolf et al., 2024; Zhang et al., 2024; Han et al., 2025). Apart from the 13S-PC identified by all groups and the 13S+ $\beta 5$ + $\beta 6$ -PC identified by Schnell et al. and Adolf et al., we also resolved the 13S+ $\beta 5$ -PC, 13S+ $\beta 1$ -PC, 13S+ $\beta 1$ + $\beta 5$ -PC, as well as the 15S-PC. Yeast complexes

elucidated by several groups are all practically identical at c-alpha level, as suggested by the RMSDs presented in the manuscript (line 127 and line 147). We cannot identify any incompatible features.

(3) The local EM densities of S3/S4 beta-hairpin loops and Pba1 loop that inserts into the alpha pocket must be presented in either main figures or supplementary figures to support the authors' claim that they indeed solved these loops to high enough resolution that allowed their accurate atomic modeling and conclusion. Similarly, the local EM densities of the N-terminal of alpha1 must be presented for visual assessment to support their atomic and functional interpretation. More generally, any "previously unrecognized details" must be supported or justified by clear presentations of high-resolution EM densities in either main figures or supplementary figures.

Our Response: Addressing the reviewers request for visual evidence, we have incorporated local EM densities into our figures: the S3/S4- β -hairpin loops are shown in Figure S4, the Pba1 loop in Extended Figure 5C, and the N-terminus of α 1 in Figure S7.

(4) In Table S1, why is the percentage of disallowed residues all above 90%? This is either done wrong, or the table contains typo.

Our Response: This was a typo. Thank you for pointing it out.

(5) The introduction could be improved in the clarification of what CP assembly intermediates have been structurally solved, what is yet to be solved, and why the present study is important. The discussion section may also be improved with a summary on the key differences between the newly solved structures and the previously published CP assembly intermediate structures.

Our Response: We revised the introduction to add more clarity with regard to already published PC structures. Specifically, we now mention all PC structures that overlap with our study (lines 58-60 and 74-78), cite further relevant publications and highlight the novelty of the structures presented here (lines 96-98). As mentioned above, we also included a new table (Table S1) enabling comparison of overlaps and novelties between already published PCs and structures presented in our manuscript. The C-alpha RMSD values between our structures and already published PCs from yeast are given in the manuscript (lines 127 and line 147). We revised the discussion section according to the reviewer's suggestion to highlight our main novel findings described in our paper, which is supported by the new figure 6 (as outlined below in our response to item 6).

(6) It would be very helpful to add a new main figure to summarize the main findings on the assembly mechanism. The logic flow of the paper is a little bit brittle and lacks focus of a central story. This may be improved with better mechanistic discussion and conclusion paragraphs at the end of the paper.

Our Response: We added a new figure (Figure 6) to provide a conceptual summary of the key findings of our study, highlighting the alternative assembly pathways, the gradual folding of Ump1, and the importance of the α 1 N-terminus for Pba1-Pba2 release, which is associated with loosening of the so far uncharacterised loop in Pba1 (residues 81-117) intercalated between α 3 and α 4. This figure includes, as indicated in the legend, six structures of yeast 13-15S-PCs, four of which have not been described before, including the first structure of a 15S-PC (half-CP without β 7; resolution of 3.1Å). In detail the figure summarizes 1) that, aside from the previously reported importance of the Pba1 and Pba2 C-termini and the Pba1 N-terminus, we identified a loop in Pba1 (residues 81-117) as a structural feature that is transiently intercalating between α 3 and α 4. 2) Furthermore, we discovered a role of the N-terminus of α 1 in the control of Pba1-Pba2 release. 3) We observe that β 1 has the capacity to be incorporated before β 5 and β 6, even though the quantity of observed complexes suggests that β 5 is more often the first subunit to be added to 13S-PCs, as is emphasized by the stronger arrows in the bottom pathway. 4) and 5) finally, as indicated by the overarching arrow at the bottom, we characterized the stepwise structural maturation of Ump1 and of the S3/S4 β -hairpin loops in the β -subunits, which prepare for final maturation upon β 7-driven PC dimerization.

Minor issues:

(1) Please present gel filtration or size-exclusion chromatographic results of the purified complexes in supplementary figures.

Our Response: We are unable to provide gel filtration data due to the limitations imposed by extensive sample dilution, which reduces protein concentration below the threshold for reliable detection. Instead we have utilized native gel electrophoresis (Figure S1) as a more sensitive and less disruptive method for analyzing the assembly intermediates.

(2) There are numerous places (e.g. line 70, 73, 74, 109, 114, etc.) where Greek alpha is written as “a”, and beta is written as “b”.

Our Response: Thank you for pointing out these inconsistencies. We corrected the typos.

(3) Table S1 may have to be moved to main text as a main Table.

Our Response: We have moved table S1 to the main section (now named Table 1).

Point-by-Point Response to Reviewer Comments

Reviewer #1 (Remarks to the Author):

In this revised manuscript, Mark et al. included additional data to address a number of concerns raised in the initial round of review, with the foremost being that the Beta1-containing 13S precursor complex was not convincingly shown to be an "on-pathway" assembly product, rather than an artifactual dissociation product or an artifact of limiting amounts of assembly chaperones that might result in altered subunit compositions. The authors have addressed this by overproducing Pba1-2 to prevent their depletion, and have demonstrated the ability to detect Beta1 in yeast cells expressing vanishingly low levels of Beta5 or Beta6, which have traditionally been considered to incorporate prior to Beta1. Although this data does not unequivocally prove a Beta1-first assembly mechanism, it does help to further substantiate this possibility and reduce the likelihood of alternative explanations. Text changes to the abstract and main article have softened this claim to a level this Reviewer believes is appropriate.

Some minor edits are noted below:

1. pg. 2, line 57: the term ortholog might be a better word choice than alias for describing the Ump1 ortholog in humans.
2. Pg. 4, line 148: the word "too" should be "to".

Our response:

We thank the reviewer for the positive assessment of our revised manuscript. Thank you for pointing out the editorial improvements, which we have made as suggested.

Reviewer #2 (Remarks to the Author):

The revised manuscript has sufficiently addressed all my questions. I recommend its publication in the present form.